

# Solvable models for 2+1D quantum critical points: Loop soups of 1+1D conformal field theories

Amin Moharramipour[1], Dan Sehayek[2] and Thomas Scaffidi[3,1]

**1** Department of Physics, University of Toronto,
60 St. George Street, Toronto, Ontario, M5S 1A7, Canada
**2** Department of Physics, University of California, Santa Barbara, California 93106, USA
**3** Department of Physics and Astronomy, University of California,
Irvine, California 92697, USA

## Abstract

We construct a class of solvable models for 2+1D quantum critical points by attaching 1+1D conformal field theories (CFTs) to fluctuating domain walls forming a "loop soup". Specifically, our local Hamiltonian attaches gapless spin chains to the domain walls of a triangular lattice Ising antiferromagnet. The macroscopic degeneracy between antiferromagnetic configurations is split by the Casimir energy of each decorating CFT, which is usually negative and thus favors a short loop phase with a finite gap. However, we found a set of 1D CFT Hamiltonians for which the Casimir energy is effectively positive, making it favorable for domain walls to coalesce into a single "snake" which is macroscopically long and thus hosts a CFT with a vanishing gap. The snake configurations are geometrical objects also known as fully-packed self-avoiding walks or Hamiltonian walks which are described by an O($n = 0$) loop ensemble with a non-unitary 2+0D CFT description. Combining this description with the 1+1D decoration CFT, we obtain a 2+1D theory with unusual critical exponents and entanglement properties. Regarding the latter, we show that the log contributions from the decoration CFTs conspire with the spatial distribution of loops crossing the entanglement cut to generate a "non-local area law". Our predictions are verified by Monte Carlo simulations.

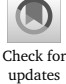

# 1  Introduction

Phases of matter can be distinguished by the presence or absence of a gap to bulk excitations. Gapped phases of matter were shown over the recent years to have an extremely rich classification of (symmetry-protected) topological (SPT) phases [1–11]. By contrast, gapless systems are often more challenging to study theoretically, with the notable exception of 1-dimensional ($D = 1$) systems about which much is known thanks to conformal field theory [12]. Further, the notion of gapless SPT order [13–24] is still an active area of research.

For gapped topological systems like SPTs, the decorated domain wall construction provides a way of constructing a $D$-dim SPT starting from a $(D-1)$-dim SPT [25]. The idea is to decorate domain walls of a (typically $G = Z_2$) symmetry with a $(D-1)$-dim SPT, and to let the domain walls fluctuate in order to restore $G$. Is it possible to generalize this dimensional "bootstrapping" approach to gapless systems? In this work, we answer this question affirmatively by providing a decorated domain wall construction in which 1+1D CFTs are attached on domain walls, leading to a gapless 2+1D theory with remarkable properties.

The idea of attaching gapless 1D theories on fluctuating domain walls is motivated by a number of physical systems. First, the combination of frustrated magnetism and itinerant degrees of freedom appears in so-called "charge-ice" systems [26, 27]. Second, decorated domain wall models were shown to be relevant to certain models combining spin and orbital degrees of freedom [28], including the extended Hubbard model on the Kagome [29] and checkerboard [30] lattice. Third, when a two-dimensional antiferromagnetic insulator is doped, the charge concentrates into domain walls forming "metallic rivers" and effectively behaving as Luttinger liquids in an active environment [31–33]. Fourth, if we reintepret the domains as distinct gapped topological phases, the gapless degrees of freedom appearing at the domain walls would have a natural intepretation as edge modes, like in the quantum Hall

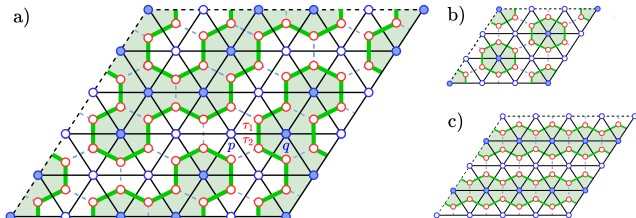

Figure 1: Example of a decorated domain wall configuration. $\sigma_z = \pm 1$ spins are represented by filled and empty blue dots, respectively. $\tau$ spins (red dots) are located on every site of the hexagonal lattice (dashed blue lines), including the domain walls of the $\sigma_z$ spins (green lines). As an example, $\tau_1$ and $\tau_2$ are located on the domain wall between sites $p$ and $q$. Figure a) illustrates a snake configuration, consisting of a single domain wall traversing the entire lattice. Figures b) and c) illustrate hexagon solid and stripe configurations, consisting of contractible and non-contractible domain walls of the shortest length, respectively.

plateau transition [34]. Finally, since Ising domain walls have been proposed as a model for strings [35–38], the decoration described here is analogous to the fermionic degrees of freedom which are added to form super-strings [39].

From a spin-liquid perspective, we will see that our construction starts with a familiar route, in which local frustration (in our case Ising antiferromagnetism on a triangular lattice) leads to an exponentially large number of classical ground states which is described by a familiar ensemble of fully packed loops (FPL) [40–42]. However, the source of the exotic criticality we will describe below — fully packed self-avoiding walks described by a non-unitary $c = -1$ theory — lies elsewhere: It is due to an additional kind of *non-local* frustration whereby all allowed loops on the honeycomb lattice realize an effectively twisted boundary condition [43] for the CFT which lives on it.

## 2 Model and solution

We introduce a local model which attaches any translation-invariant 1D Hamiltonian $H_{1D}$ to Ising domain walls. The model contains $\sigma$ Ising spins living on a triangular lattice, and $\tau$ decoration degrees of freedom (DOFs) living on the vertices of the dual honeycomb lattice (see Fig. 1). The $\tau$ operators could be anything (spins, bosons, fermions,...), but we will take them to be spins for concreteness. The Hamiltonian couples the $\tau$ DOFs only along the domain walls of the $\sigma$ spins. This is easily implemented on this lattice, as we now show for an example which involves nearest-neighbor coupling terms $H_{\text{dec}}(\tau_1, \tau_2)$ between $\tau$ DOFs (e.g. $H_{\text{dec}}(\tau_1, \tau_2) = \tau_1^x \tau_2^x + \tau_1^y \tau_2^y + \Delta \tau_1^z \tau_2^z$ for the XXZ chain):[1]

$$H = \sum_{\langle pq \rangle} H_{pq} = \sum_{\langle pq \rangle} \frac{1 - \sigma_p^z \sigma_q^z}{2} H_{\text{dec}}(\tau_1^{pq}, \tau_2^{pq}), \tag{1}$$

where $\langle pq \rangle$ denotes bonds of the triangular lattice and $\tau_1^{pq}$ and $\tau_2^{pq}$ are the spins attached to the honeycomb bond crossing $\langle pq \rangle$ (see Fig. 1). One can easily generalize this construction to the case of $H_{\text{dec}}$ with longer-range terms, as shown in Appendix A.

---

[1]This example assumes $H_{\text{dec}}(\tau_1, \tau_2) = H_{\text{dec}}(\tau_2, \tau_1)$ for simplicity. If the 1D Hamitonian is not inversion-symmetric, one needs to have a consistent way of orienting the domain walls. This is easily done by choosing a convention such as: when moving from the up spins to the down spin domain, the domain wall is oriented to the left.

The Hamiltonian 1 is block diagonalized in the $\sigma^z$ basis. There is a two to one mapping between $\{\sigma^z\}$ configurations and their domain wall configurations, which form a set of non-crossing loops $\mathcal{L}$. This enables us to rewrite the Hamiltonian as a sum of 1D Hamiltonians living on each loop $l$:

$$H = \sum_{\{\sigma^z\}} \sum_{l \in \mathcal{L}} H_{1D}[l], \tag{2}$$

where $H_{1D} = \sum_i H_{dec}(\tau_i, \tau_{i+1})$ is the 1D Hamiltonian given by the sum of the $H_{dec}$ terms along the loop. Each $H$ eigenstate, denoted $|\Psi\rangle$, is a tensor product of an $H_{1D}$ eigenstate $|\psi_{1D}\rangle$ on each loop $l$. We will be mostly interested in the case for which $|\psi_{1D}\rangle$ is the ground state of $H_{1D}$, given by:[2]

$$|\Psi[\mathcal{L}]\rangle = |\{\sigma^z\}\rangle \bigotimes_{l \in \mathcal{L}} \left|\psi_{1D,GS}[l]\right\rangle. \tag{3}$$

The total energy of this state is the sum of the 1D ground state energies, $\mathcal{E}[\mathcal{L}] = \sum_{l \in \mathcal{L}} E_{GS}(L_l)$ with $H_{1D}[l]\left|\psi_{1D,GS}[l]\right\rangle = E_{GS}(L_l)\left|\psi_{1D,GS}[l]\right\rangle$ and where $L_l$ is the length of loop $l$.

Which configurations $\mathcal{L}$ minimize the total energy $\mathcal{E}[\mathcal{L}]$? Let us for now neglect finite-length effects in $E_{GS}(L)$ and only keep the leading-order term in $L$: $E_{GS}(L) \simeq \epsilon_0 L$, with $\epsilon_0$ the ground state energy per site. If $\epsilon_0 < 0$, minimizing the energy simply amounts to maximizing the total length of all domain walls, which corresponds to a "fully packed" loop configuration for which each honeycomb site is visited by a loop. The corresponding $\{\sigma^z\}$ are the ground states of the classical Ising AFM on a triangular lattice, first studied by Wannier. These configurations lead to a residual entropy per spin of 0.323066 [44]. Note that $\epsilon_0$ can always be made negative by adding a term $-J\mathbb{1}$ to $H_{dec}$ with a large enough $J$. For the sake of simplicity, we will work in the limit of $J \to \infty$ for the remainder of this work, which means the only allowed loop configurations are fully packed.

At this point, we have an exponential degeneracy between FPL configurations. However, this degeneracy will in general be split by the finite-length corrections to the ground state energy of each chain. For the rest of the letter, we will focus on the case when $H_{1D}$ realizes a conformal field theory (CFT). For a CFT with periodic boundary conditions, the finite-size correction to the ground state energy — also called the Casimir energy — is expected to be universal and proportional to the central charge $c$: $E_{GS}/L = \epsilon_0 - \pi c/3L^2 + \dots$[3] Naively, this should put a damp on our hopes of realizing a gapless theory: since $c > 0$ for unitary theories, this means the Casimir energy is negative and the energy per site is thus minimized for short loops. In this scenario, the lowest-energy loop configurations are thus the ones which pave the plane with the shortest possible loops (which are hexagons of length 6), leading to a 3-fold breaking of lattice translation in a "hexagon solid" phase (see Fig. 1.b) [29]. Since each loop has finite size, the 1D Hamiltonian living on it has a finite gap, and the theory is thus gapped.

However, if $c$ was negative, the energy per site would be minimized by taking $L \to \infty$, leading to a loop which is macroscopically long and which can thus host a 1+1D theory with a vanishing gap. Interestingly, this scenario can be realized by choosing CFTs which are "frustrated" for certain chain lengths [43] and which can effectively behave as if $c < 0$, as we now show. A key insight is that, for certain CFTs, the value of $c$ which appears in the Casimir energy is not always equal to the actual central charge and may depend on the chain length modulo some integer. We will focus on cases where that integer is 4, leading to a more general formula

---

[2]We have omitted in Eq. 3 the $\tau$ DOFs living on empty sites (i.e. honeycomb sites not traversed by a loop), which form decoupled zero modes since they trivially commute with the Hamiltonian. The reason is that we will soon restrict the discussion to fully-packed configurations, which do not have empty sites.

[3]We use a convention in which the CFT velocity is 2.

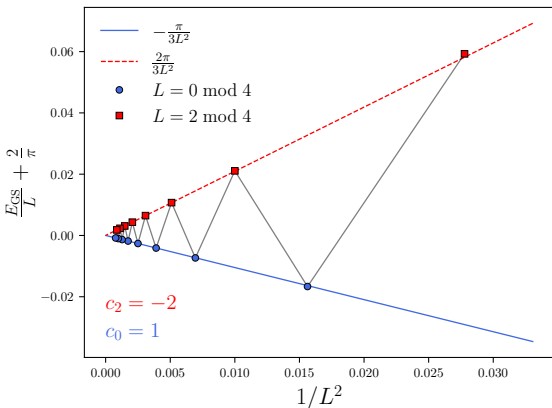

Figure 2: The energy density of the ground state of the $H_{\text{X-ZXZ}}$ chain as a function of its length $L$ (markers) along with a fit to Eq. 4 for the $c_0 = 1$ and $c_2 = -2$ branches (curves). Specifically, the figure shows the graph of $E_{\text{GS}}/L + \epsilon_0$ vs $1/L^2$, where the thermodynamic limit value for the energy density of $\epsilon_0 = 2/\pi$ was used. Although higher-order terms in $1/L$ could in principle become sizable for short loops, we find that the $-\pi c/3L^2$ formula (curves) for the energy density agrees extremely well with the exact values (markers) down to $L = 6$, which corresponds to the shortest loop allowed by the honeycomb lattice.

for the finite-size ground state energy density:[4]

$$\frac{E_{\text{GS}}(L = 4k + r)}{L} = \epsilon_0 - \frac{\pi c_r}{3}\frac{1}{L^2} + \dots \tag{4}$$

For example, consider the following 1D Hamiltonian, which is the decoration model we will focus on for the rest of the article:[5]

$$H_{\text{X-ZXZ}}(L) = \frac{1}{2}\sum_{i=1}^{L}(\tau_i^x - \tau_{i-1}^z \tau_i^x \tau_{i+1}^z). \tag{5}$$

It describes the quantum phase transition between a 1D trivial paramagnet and an SPT protected by a $Z_2 \times Z_2$ symmetry which is generated by $\prod_i \tau_{2i}^x$ and $\prod_i \tau_{2i+1}^x$ [45].[6,7] The mod 4 effect in the energy density of Eq. 5 can be calculated analytically (see Appendix B) and gives $c_0 = 1$ and $c_2 = -2$ (see Fig. 2).[8] The most relevant aspect for us will be that $c_2 < 0$. Another example of a CFT with $c_2 < 0$ is the doubled version of the XX chain with

---

[4]One might worry that this CFT formula only holds for long loops, but we have found that it actually gives extremely good agreement with exact values down to the shortest loops possible (which have $L = 6$ on the honeycomb lattice), see Fig. 2. More generally, the precise value of the energy density for short loops does not really matter so long as the energy density is a monotonically decreasing function of $L$ for $L = 4k + 2$, which we have found to be the case for all models we considered (see Fig. 2 and Appendix C).

[5]Note that this 1D Hamiltonian has single and three-body terms, whereas the example given in Eq. 1 was for two-body terms. A generalization of Eq. 1 for single and three-body terms is given in Appendix A.

[6]A quantum phase transition with the same universality also could also appear in spin-1 chain models which are relevant for a number of materials like $CsNiCl_3$ [46] and $Y_2BaNiO_5$ [47], in which case it separates the Haldane phase [48] from a topologically trivial phase. We discuss this spin-1 model in Appendix C.

[7]We note that Eq. 5 also describes the edge theory of a 2D $Z_2$ SPT [9].

[8]This mod 4 effect can be understood at a simple level: as explained in Appendix B, the model can be solved by performing a Jordan-Wigner whereby the Ising domain walls become fermions. Because the number of domain walls is restricted to be even for periodic boundary conditions, so is the number of fermions. This means only chains of $L = 4k$ can be at exactly half-filling, and the other ones are thus "frustrated" and have a higher energy density.

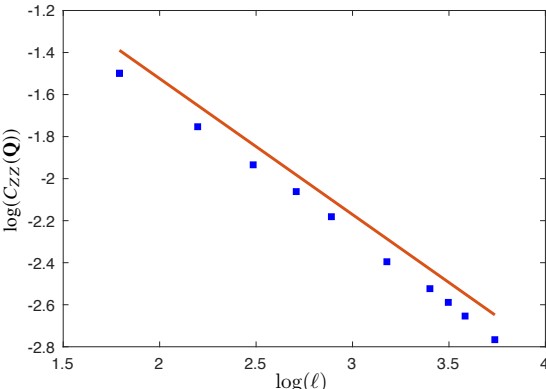

Figure 3: Peak value of equal-time structure factor $C_{ZZ}(\mathbf{Q}) \equiv \ell^{-2}\sum_{\mathbf{x}}C_{ZZ}(\mathbf{x})e^{-i\mathbf{Q}\cdot\mathbf{x}} \sim \ell^{-\eta}$ where $\mathbf{Q}$ is at the corner of the Brillouin zone (see also Appendix. E.1 for more details). The linear fit gives $\eta = 0.65 \pm 0.01$. The calculations have been performed at $T\ell^2 \equiv \tilde{T} = 4$.

$H_{1D} = \frac{1}{2}\sum_{i=1}^{L}(\tau_i^x\tau_{i+2}^x+\tau_i^y\tau_{i+2}^y)$ which has $c_0 = 4$ and $c_2 = -11$[9] (see Appendix C for a detailed calculation and for more examples of decoration Hamiltonians).

Another crucial insight is that, on the honeycomb lattice, all contractible loops forming an FPL configuration have a length given by $L = 4k + 2$ with $k$ an integer (see Appendix D for a proof). Putting aside non-contractible loops for now, this means the low energy properties of the system are only determined by the sign of $c_2$. For $c_2 > 0$, the system forms a gapped solid of short loops, as explained before. However, for $c_2 < 0$, the Casimir energy is positive and thus the minimal energy per site is obtained for $L \rightarrow \infty$. For a 2D system of linear size $\ell$, the maximal loop length scales like $\ell^2$ and is obtained for a single loop visiting every site of the lattice exactly once (also called a Hamiltonian walk, or fully-packed self-avoiding walk (FPSAW), or "snake" in the rest of the paper). Since the gap $\Delta$ of $H_{1D}$ scales like the inverse length $L$ of the chain on which it lives, one finds $\Delta \sim 1/L \sim 1/\ell^2$, which indicates a 2D gapless theory with dynamical critical exponent $z = 2$.

Since there is an extensive number of degenerate "snake" configurations (with entropy per site $s \equiv S/N \approx 0.130812$) [49], the low-energy physics is described by a statistical average over them, which is obtained in the zero-$T$ limit of the following thermal density matrix:

$$\rho = \sum_{\{\sigma^z\}} p_{\{\sigma^z\}} |\{\sigma^z\}\rangle \langle\{\sigma^z\}| \bigotimes_{l\in\mathcal{L}} \rho_{1D}[l], \tag{6}$$

where $\rho_{1D}[l] = \exp(-\beta H_{1D}[l])/Z(L_l)$ with $Z(L_l) = \text{Tr}[\exp(-\beta H_{1D}[l])]$, and where $p_{\{\sigma^z\}} = \prod_{l\in\mathcal{L}} Z(L_l)/Z$ with $Z = \sum_{\{\sigma^z\}}\prod_{l\in\mathcal{L}} Z(L_l)$. In the zero-$T$ limit,[10] the ensemble of $\{\sigma^z\}$ described by $\rho$ maps to the O($n \rightarrow 0$) fully packed loop model, which is described by a $c = -1$ non-unitary CFT [49]. This leads to unusual power laws for correlation functions. For example, the antiferromagnetic correlations captured by $C_{ZZ}(\mathbf{x}) = \langle\sigma^z(0)\sigma^z(\mathbf{x})\rangle \equiv Z^{-1}\text{Tr}[\rho\sigma^z(0)\sigma^z(\mathbf{x})]$ have a power law envelope $C_{ZZ}(\mathbf{x}) \sim |\mathbf{x}|^{-\eta}$, which we extract from the structure factor (see Fig. 3 and Appendix. E.1). We find $\eta = 0.65 \pm 1$, which agrees well with the prediction of $\eta = 2/3$ based on a Coulomb gas description of the $c = -1$ CFT (see Appendix E.2). This also shows conclusively that the snake phase is in a different universality class from the standard Ising triangular lattice antiferromagnet (also known as the O($n = 1$) fully-packed loop model) which has $\eta = 0.5$ [50].

---

[9]We conventionally define $L$ as the number of sites, not the number of unit cells. Thus, we obtain $c_0 = 4$ instead of the expected $c_0 = 2$ for two independent chains of $c_0 = 1$.

[10]The zero-$T$ limit needs to be taken after the thermodynamic limit, as discussed below.

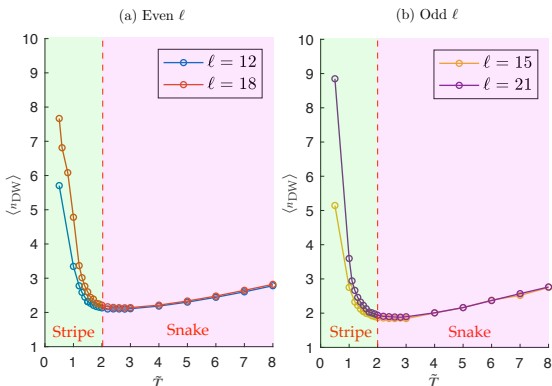

Figure 4: The average number of domain walls $\langle n_{DW} \rangle$ vs rescaled temperature $\tilde{T} = T\ell^2$, for an $\ell$ by $\ell$ torus. (Due to an even-odd effect, the data for even and odd $\ell$ was separated into two panels for clarity.) The plot indicates a phase transition between a snake phase at high $\tilde{T}$ for which $\langle n_{DW} \rangle$ does not scale with $\ell$, and a stripe phase at low $\tilde{T}$ for which $\langle n_{DW} \rangle$ grows with $\ell$.

We now discuss correlation functions of $\tau$ DOFs. Denoting $\phi(\mathbf{x}, t)$ a scaling operator in the decoration CFT with scaling dimension $\Delta_\phi$, purely temporal correlations have the same value for any snake configuration, leading to:

$$\langle \phi(\mathbf{0}, 0)\phi(\mathbf{0}, t) \rangle \sim 1/|t|^{2\Delta_\phi} . \tag{7}$$

By contrast, for spatial correlations the average over snake configurations leads to an averaging over the 1D distance $d$ measured along the snake which appears in the correlator $\sim 1/|d|^{2\Delta_\phi}$. Assuming $\phi$ is chosen so that it has no lattice-scale oscillations, we expect it is safe to replace $|d|$ by its average value, which scales like $|d| \sim |\mathbf{x}|^{1/\nu}$, with $\nu = 1/2$ the known geometrical exponent of the FPSAWs. This leads to the following prediction for spatial correlations:

$$\langle \phi(\mathbf{0}, 0)\phi(\mathbf{x}, 0) \rangle \sim 1/|d|^{2\Delta_\phi} \sim 1/|\mathbf{x}|^{4\Delta_\phi} . \tag{8}$$

To summarize, temporal scaling dimensions in the 2+1D theory are the same as for the underlying decoration 1+1D CFT, whereas spatial ones are multiplied by 2. This behavior is manifestly consistent with $z = 2$. (It is instructive to contrast this behavior with conformal quantum critical points (CQCPs) [51–53], which are 2+1D critical points constructed starting from a 2+0D CFT, and which often have $z = 2$. For CQCPs, the spatial scaling dimensions are the same as the underlying 2+0D CFT, whereas the dynamical ones are divided by 2.)

## 3 Finite temperature and finite size

How do the previous results survive at finite temperature? In order to study this, we have developed a worm Monte Carlo algorithm [54–56] (see Appendix F for more details) which probes the density matrix in Eq. 6. We calculated the average number of domain walls $\langle n_{DW} \rangle$ as a function of temperature (see Fig. 4), from which we can also infer the typical length of each domain wall $L \sim \ell^2 / \langle n_{DW} \rangle$. We find that, in the snake phase, the average number of domain walls scales as $\langle n_{DW} \rangle \sim \ell^2/\beta$, and the typical length of each domain wall therefore scales as $L \sim \beta$. Finite temperature thus provides a spatial infrared cutoff of size $\beta$ for the chain lengths on which the 1+1D CFTs live, along with the usual temporal cutoff $\beta$ in imaginary time. This means that, as $T$ approaches zero, both the spatial and temporal dimensions of the 1+1D tori

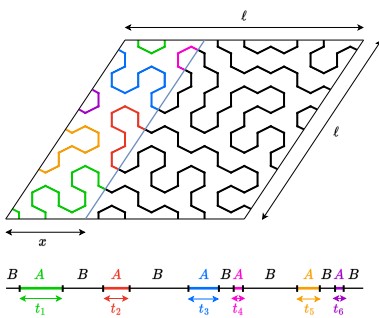

Figure 5: Top: Partition of an $\ell$ by $\ell$ torus for which we calculate entanglement. Bottom: After unfolding the snake, the partition becomes a union of disjoint intervals. Intervals belonging to the subsystem $A$ are colored while those outside $A$ are black and labeled $B$.

hosting the CFTs diverge like $\beta$, in contrast to usual quantum critical scaling for which only the imaginary time dimension scales with $\beta$ whereas the spatial dimension is independent of $\beta$.

While the discussion so far was done in the limit of $\ell \to \infty$ before $T \to 0$, let us now consider a finite-size system, focusing on an $\ell$ by $\ell$ torus for concreteness. The main new ingredient on the torus is the presence of non-contractible loops. Indeed, the geometrical constraint that loop lengths have $L = 4k+2$ for FPL configurations only applies to contractible loops. This means non-contractible loops can have $L = 4k$, and can thus be on the $c_0 > 0$ branch, which is lower in energy (see Fig. 2). Filling the space with such loops amounts to forming a stripe configuration for the $\sigma^z$ spins (see Fig. 1.c). The difference in total energy between the snake and the stripe configuration is calculated easily: $\Delta\mathcal{E} = \frac{\pi c_0}{6} + O(1/\ell^2)$. Since $c_0 > 0$, the stripe phase is lower in energy, and is actually the true ground state for a finite-size torus. However, the snake phase has a zero-temperature finite entropy density $s$, whereas the stripe phase does not. Any small temperature should thus be enough to stabilize the snake phase at the expense of the stripe phase. Let us consider the difference in free energy between the two phases, given by $\Delta F = \Delta\mathcal{E} - 2T\ell^2 s$. Since $\Delta\mathcal{E} = \mathcal{O}(1)$, the entropy term dominates already at a temperature which is parametrically small in system size, motivating the definition of a rescaled temperature $\tilde{T} \equiv T\ell^2$. Setting $\Delta F = 0$, we predict a first-order phase transition[11] at $\tilde{T}_c = \pi c_0/12s$ between a stripe phase at low $\tilde{T}$ and a snake phase at higher $\tilde{T}$. This gives $\tilde{T}_c \simeq 2.0013$ for the decoration Hamiltonian $H_{\text{X-ZXZ}}$ with $c_0 = 1$, which is confirmed by our numerics (see Fig. 4). For $\tilde{T} > \tilde{T}_c$, we observe that the average number of domain walls is independent of system size, consistently with the snake phase. For $\tilde{T} < \tilde{T}_c$, $\langle n_{\text{DW}} \rangle$ shows an increase with $\ell$, consistently with a stripe phase.

In practice, since numerics is done for finite $\ell$, we have to work at $\tilde{T} > \tilde{T}_c$ in order to probe the snake phase, for which $\langle n_{\text{DW}} \rangle$ is not strictly equal to one, but remains of order $O(1)$ (for example, we used $\tilde{T} = 4$ to generate data for Fig. 3, for which $\langle n_{\text{DW}} \rangle$ is slightly above 2). At any rate, local properties should not be able to distinguish between a single or an $O(1)$ number of snakes.

---

[11]Since that transition would break simultaneously the Ising symmetry of the $\sigma$ spins and the rotational symmetry of the lattice, it needs to be first-order [56]. Another possibility is the existence of an intermediate nematic phase, which we were not able to observe with our numerics.

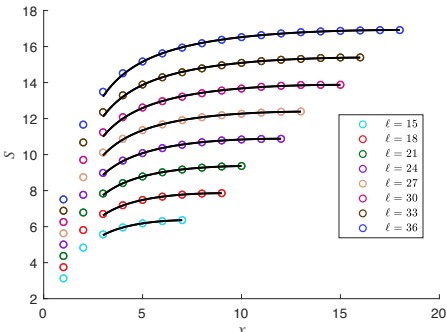

Figure 6: Universal contributions to the entanglement entropy of the strip shown in Fig. 5 vs. $x$ for different system sizes $\ell$. The black curves are fitted to $x \geq 4$ numerical data points according to Eq. 9. The snake configurations were generated at $\tilde{T} = 4$. Details on how $S$ was calculated numerically, and about the non-universal contributions which were dropped, can be found in Appendix G.

## 4 Entanglement

Should we expect an $\ell_A \log(\ell_A)$ term (with $\ell_A$ the linear size of subsystem A) in the bipartite entanglement of a 2D soup of 1+1D CFTs such as the one we have constructed here (as suggested in Ref. [57] for example)? Surprisingly, we find that this is not the case, due to a subtle property of the spatial distribution of loops crossing the entanglement cut which restores the area law in 2D.[12] (It is however a non-local form of the area law, as we will explain).

Let us first give a physical argument for the restoration of the area law, followed by numerical results. As shown in Fig. 5, for any given snake configuration $\mathcal{L}$, unfolding the snake to a straight line results in a periodic chain for the $\tau$ DOFs in which the subsystem $A$ is mapped into several disjoint intervals. Hence, to determine the entanglement entropy $S[\mathcal{L}]$ of a given state $|\Psi[\mathcal{L}]\rangle$, we only need to apply the known formula for the entanglement of disjoint intervals in a 1+1D CFT [58–60] (refer to Appendix G.1 for more details). In order to derive the entanglement scaling, we assume it is sufficient to approximate this formula by $S \sim (c/3) \sum_{i=1}^{N_{\text{int}}} \log(t_i)$ where $t_i$ is the length of interval $i$ and $N_{\text{int}}$ is the number of intervals. We then average $S[\mathcal{L}]$ over $\mathcal{L}$ which, assuming a translation-invariant entanglement cut like that of Fig. 5 for simplicity, effectively corresponds to an average over interval lengths denoted by $\overline{f(t)} \equiv \int dt \, p(t) f(t)$ with a distribution $p(t_i) \equiv p(t)$. We also know that $N_{\text{int}} = 2\ell_A/3$ since each honeycomb edge crossed by the entanglement cut has a 2/3 probability of being occupied by a loop strand.[13] Overall, this leads to a simple prediction for the entanglement: $\overline{S} \propto (c/3)(2\ell_A/3)\overline{\log(t)}$.

The remaining task is thus to find how $\overline{\log(t)}$ scales with $\ell_A$. Although $p(t)$ is a priori unknown, we already know that $\overline{t} \propto \ell_A$ since the total length of all intervals should scale like $\ell_A^2$ because the snake visits every site inside subsystem $A$. If we assumed that $\overline{\log(t)} \sim \log(\overline{t})$, we would thus find a $\ell_A \log(\ell_A)$ term for the entanglement. This is incorrect however because, as we argue in Appendix G.3, $p(t)$ effectively describes the distribution of the exit time at which a random walker first exits subsystem $A$, having started inside $A$ one lattice spacing away from the entanglement cut. Such a process is expected to follow a Lévy-type distribution, for which the typical value $t_{\text{typ}}$ is $O(1)$ even though the average value $\overline{t}$ diverges with $\ell_A$ due to a long-time tail. One thus finds $\overline{\log(t)} \sim \log(t_{\text{typ}}) = O(1)$, and the area law is restored: $\overline{S} \sim \ell_A$.

---

[12]In the stripe phase, there is actually a $\ell_{A,y} \log(\ell_{A,x})$ term (for stripes parallel to the $x$ axis, with $\ell_{A,x}, \ell_{A,y}$ the dimensions of subsystem $A$ along $x$ and $y$), since one can simply add the log contribution from each stripe crossing the partition cut.

[13]Strictly speaking, $N_{\text{int}}$ can have small fluctuations away from $2\ell_A/3$ depending on the snake configuration, but these fluctuations can be neglected in the large $\ell_A$ limit we care about.

We now focus on a strip subsystem of width $x$ in a $\ell$ by $\ell$ torus (see Fig 5) [61]. Using the random walker model mentioned above, we derive a distribution $p(t)$ and calculate $\overline{\log(t)}$ as a function of $x$ (see Appendix G.3 for details), leading to the following prediction for the entanglement:

$$S_{\text{strip}}(\ell, x) = 2\ell\left(A - \frac{B}{\tilde{x}}\right) + o(\ell), \tag{9}$$

with $\tilde{x} = \frac{\ell}{\pi}\sin\left(\frac{\pi x}{\ell}\right)$. This formula gives a good agreement with our numerical results, as shown in Fig. 6 (see also Appendix G.4). In Eq. 9, the dependence of the area law prefactor on $x$ due to the term proportional to $B$ is unusual: it does not appear in the standard scaling forms used for gapless 2D systems, and leads to a much larger dependence on $x$ of $S_{\text{strip}}$ than usually observed [61, 62]. The $B$ term depends crucially on contributions from parametrically long intervals and thus reveals the non-local character of the area law: the distribution $p(t)$ has a long-time tail which extends until the "Thouless time" $t_{\text{Th}} = x^2/D$, with $D$ the effective diffusion constant of the random walker. (Another example of a non-local area law was recently proposed in Ref. [57]).

## 5 Discussion

By attaching CFTs with positive Casimir energy on domain walls, we have shown how to realize a 2+1D quantum critical point featuring a single domain wall visiting every site of the system, whose statistical fluctuations are described by the $O(n = 0)$ fully-packed loop ensemble.

On the border of which phases does this QCP exist? There are at least two types of relevant perturbations. The first type is obtained by adding terms to the decoration Hamiltonian. In that case, any relevant perturbation of the 1+1D CFT would also be relevant for the 2+1D QCP. For example, for the case of $H_{\text{X-ZXZ}}$ considered in this work, we can use the following interpolation Hamiltonian as decoration:

$$H_{\text{1D}}(\alpha) = \sum_{i=1}^{L}(1-\alpha)\tau_i^x - \alpha\tau_{i-1}^z\tau_i^x\tau_{i+1}^z, \tag{10}$$

with $0 \leq \alpha \leq 1$. In this notation, the unperturbed decoration Hamiltonian $H_{\text{X-ZXZ}}$ corresponds to $\alpha = 1/2$, at which the QCP occurs. For $\alpha < 1/2$ (resp. $\alpha > 1/2$), $H_{\text{1D}}(\alpha)$ flows to a trivial (resp. non-trivial) $Z_2 \times Z_2$ 1D gapped bosonic SPT [45].

When the decoration Hamiltonian flows to a gapped phase, we expect the spin degrees of freedom $\sigma^z$ to flow to the conventional triangular lattice Ising antiferromagnet, also known as the $O(n = 1)$ fully-packed loop ensemble [40, 41]. Indeed, in the limit of vanishing correlation length for the decoration DOFs, the ground state energy of $H_{\text{1D}}$ becomes independent of the domain wall length, and all fully-packed loop configurations become equally likely. For the example at hand, the QCP thus separates two phases of fully-packed loops with $n = 1$ fugacity which are decorated by a trivial (resp. non-trivial) 1D gapped $Z_2 \times Z_2$ SPT. (The 2D Hamiltonian obtained by using $H_{1D}(\alpha)$ as decoration corresponds to the same interpolation Hamiltonian between 2D $Z_2^3$ SPTs introduced in Ref. [63] (See also Refs. [64–68]), but with the addition of an infinitely large nearest-neighbor antiferromagnetic coupling on one of the three triangular sublattices in order to enforce the fully-packed loop constraint on the $\sigma^z$ spins). The class of QCPs we have proposed appear thus naturally at the transition between different 2D gapped SPTs with an underlying domain wall structure. Further, the kind of statistical average over snake configurations we constructed could describe a phase transition between average SPTs, which were introduced recently [69, 70].

A second type of relevant perturbation is obtained by adding terms to the Hamiltonian for the $\sigma$ degrees of freedom. The most natural term to add would be a transverse field term $\sigma^x$,

which would generate quantum fluctuations between the snake configurations. In fact, quantum fluctuations between decorated loops were studied in the context of the Kagome Hubbard model in Ref. [29]. In that work, the relevant classical loop configurations formed a solid of short loops (the "hexagon solid" pictured in Fig. 1b), and the quantum fluctuations thus naturally stabilized a "plaquette" phase of resonant short loops. In our case, the relevant classical configurations feature a single snake, and the impact of local quantum fluctuations on such a non-local object seems difficult to predict a priori. Nonetheless, the possibility of stabilizing a quantum superposition of snake configurations, in analogy with earlier work work [51,71,72], is interesting and left for future work.

We note that the snake phase could be amenable to a fermionic description after performing a Jordan-Wigner transformation on the $\tau$ DOFs along the snake. A similar construction was used to study spin liquid models like the Kitaev honeycomb model [73,74], but for a fixed snake configuration with a simple geometry which necessarily breaks certain spatial symmetries, whereas in our case the symmetry is restored by averaging over snake configurations. Once expressed in terms of fermions, the transition from the stripe phase to the snake phase would be reminiscent of the smectic and/or nematic transitions in electronic systems [75].

Our construction generates a new kind of non-local frustration by combining two primary ingredients: (1) a non-trivial dependence of the Casimir energy of the decoration CFT on the chain length modulo some integer, and (2) a geometrical constraint which forces all loops on a given lattice to have a certain length modulo some integer. The first ingredient was recently understood as arising from effectively twisted boundary conditions for certain chain lengths [43]. More generally, our work demonstrates the importance of the coefficients $c_r$ (which encode the Casimir energy dependence on the chain length modulo some integer) as an additional property of a CFT Hamiltonian with crucial physical consequences. It is remarkable that two CFTs with the same central charge can lead to completely different physics when used as decoration in our model due to their different sign for $c_2$ (e.g. the XX chain versus $H_{\text{X-ZXZ}}$). One natural generalization of our construction is to consider other lattices for which domain wall lengths are constrained in some other way, which could magnify the effect of the $c_r$ coefficients for $r \neq 2$. Another generalization is to consider other decoration Hamiltonians beyond the ones we have proposed here. This motivates further work on classifying the possible $c_r$ sequences for known CFTs, especially beyond $c = 1$ theories [43].

Finally, our calculation of the entanglement, which combines known results about entanglement in 1+1D CFTs with properties of exit time distributions, allowed us to explain the somewhat surprising presence of an area law and could be useful in other contexts, including other kinds of constrained models like that of Ref. [57]. Since the bipartite entanglement turned out to follow an area law, one wonders whether other measures of entanglement could provide a more direct probe of the non-locality arising from parametrically long intervals.

## Acknowledgments

TS acknowledges Ehud Altman, Xiangyu Cao, Sasha Chernyshev, Ryan Lanzetta, Abhinav Prem and Gabriel Wong for illuminating discussions. The DMRG calculations in this paper were completed using the ITensor package [76].

**Funding information** This research was enabled in part by support provided by Compute Canada. The authors acknowledge the hospitality of the KITP during the program "A Quantum Universe in a Crystal: Symmetry and Topology across the Correlation Spectrum". This research was supported in part by the National Science Foundation under Grant No. NSF PHY-1748958.

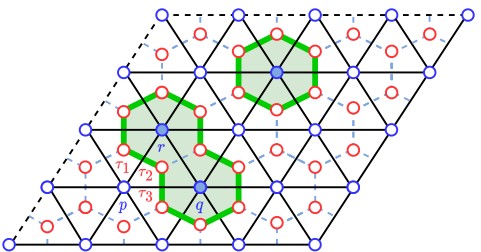

Figure 7: Example of a decorated domain wall configuration. $\sigma^z = \pm 1$ spins live on a triangular lattice and are represented by filled and empty blue dots, respectively. $\tau$ spins (red dots) are located on the vertices of the hexagonal lattice dual to the triangular lattice of the $\sigma$ spins. Domain walls of $\sigma^z$ spins are depicted as green lines. An example of a triangle of $\sigma^z$ spins $\triangle pqr$ and the corresponding $\tau$ spins is also shown in reference to Eq. A.1.

## A  Generalization of the decoration Hamiltonian to further neighbor interactions

In the main text, we showed how to construct a 2D local Hamiltonian which attaches to domain walls any 1D decoration Hamiltonian composed of nearest-neighbor terms. In this appendix we generalize this construction to terms involving three consecutive sites, using for concreteness the example of the $H_{\text{X-ZXZ}}$ Hamiltonian we consider in the main text. The same procedure can be used for terms involving any number of sites.

For the X-ZXZ model given in Eq. 5, $H_{\text{dec}}(\tau_{i-1}, \tau_i, \tau_{i+1}) = \frac{1}{2}(\tau_i^x - \tau_{i-1}^z \tau_i^x \tau_{i+1}^z)$ should be attached to domain walls of $\sigma$ spins. Since $H_{\text{dec}}$ involves three honeycomb vertices and correspondingly two honeycomb bonds, we need two domain wall projectors to modify Eq. 1:

$$H = \sum_{\triangle_{pqr}} \sum_{\substack{b_1 = \langle p_1 p_2 \rangle \in \triangle_{pqr} \\ b_2 = \langle p_3 p_4 \rangle \in \triangle_{pqr} \\ b_1 \neq b_2}} \frac{1 - \sigma_{p_1}^z \sigma_{p_2}^z}{2} \frac{1 - \sigma_{p_3}^z \sigma_{p_4}^z}{2} H_{\text{dec}}\left(\tau_1^{b_1, b_2}, \tau_2^{b_1, b_2}, \tau_3^{b_1, b_2}\right), \tag{A.1}$$

where the first sum is over triangular plaquettes of the triangular lattice, the second sum is a sum over the three choices of pairs of edges $b_1 \neq b_2$ belonging to the triangle $\triangle_{pqr}$, and where $\tau_{1,2,3}^{b_1, b_2}$ are the three $\tau$ spins on the honeycomb vertices which are connected by the honeycomb bonds dual to $b_1$ and $b_2$ (see Fig. 7 for an example).

## B  Solution of the X-ZXZ model

In this section, we review the mapping of the $H = X - ZXZ$ Hamiltonian to free fermions. We begin with a Kramers-Wannier duality transformation, for which it is convenient to split the Hilbert space into two sectors: $\prod_i \tau_i^x = \pm 1$. The constraint for the positive sector can be satisfied with a standard duality transformation: $\tilde{\tau}_i^z = \tau_i^z \tau_{i+1}^z$ and $\tau_i^x = \tilde{\tau}_{i-1}^x \tilde{\tau}_i^x$ for all $i$. Then it follows that Eq. 5 maps to

$$H_+ = \frac{1}{2} \sum_{i=1}^{L} \tilde{\tau}_i^x \tilde{\tau}_{i+1}^x + \tilde{\tau}_i^y \tilde{\tau}_{i+1}^y. \tag{B.1}$$

For the negative sector, we employ the same mapping with the modification $\tau_1^x = -\tilde{\tau}_L^x \tilde{\tau}_1^x$ to satisfy the $\prod_i \tau_i^x = -1$ constraint. Then Eq. 5 maps to

$$H_- = \frac{1}{2} \sum_{i \le L-1} \tilde{\tau}_i^x \tilde{\tau}_{i+1}^x + \tilde{\tau}_i^y \tilde{\tau}_{i+1}^y - \frac{1}{2} \left( \tilde{\tau}_L^x \tilde{\tau}_1^x + \tilde{\tau}_L^y \tilde{\tau}_1^y \right). \tag{B.2}$$

$H_+$ and $H_-$ can then both be mapped to free fermions using a Jordan-Wigner transformation

$$H_\pm = -\sum_{i < L-1} \left( c_i^\dagger c_{i+1} + \text{hc} \right) \mp s \left( c_L^\dagger c_1 + \text{hc} \right), \tag{B.3}$$

where $s = (-1)^{L-N+1}$ for $L$ sites and $N$ particles. In this free fermion language, ground state energies can be easily computed. In the case of even $L$, $s = -1$ for both sectors, since the number of domain walls $N$ must always be even. Hence, for even $L$, the positive and negative sectors possess antiperiodic and periodic boundary conditions, respectively.

For a chain of length $L = 0 \mod 4$, the ground state occurs in the positive sector with half-filling $N = L/2$. The corresponding energy density is given by

$$E_{\text{GS}}/L = -\frac{2}{L} \sum_{n=-L/4}^{L/4-1} \cos \left( 2\pi \left( n + \frac{1}{2} \right) / L \right), \tag{B.4}$$

since the momentum is quantized as $k = 2\pi \left( n + \frac{1}{2} \right)/L$ in the case where $L - N$ is even. Eq. B.4 can be expressed in closed form as $E_{\text{GS}}/L = -\frac{2}{L} \frac{1}{\sin(\pi/L)}$. For a chain of length $L = 2 \mod 4$, we are restricted from $N = L/2$ since the number of particles must be even. Hence, the ground state occurs in the positive sector with $N = L/2 \pm 1$. These two degenerate ground states have an energy density given by

$$E_{\text{GS}}/L = -\frac{2}{L} \sum_{n=-(L-2)/4}^{(L-2)/4-1} \cos \left( 2\pi \left( n + \frac{1}{2} \right) / L \right), \tag{B.5}$$

which can be expressed in closed form as $E_{\text{GS}}/L = \frac{-2}{L} \cot \left( \frac{\pi}{L} \right)$. The Casimir energy then can be obtained from these closed form expressions in the limit of large $L$:

$$\frac{E_{\text{GS}}(L)}{L} = \begin{cases} -\frac{2}{L \sin(\pi/L)} \simeq -\frac{2}{\pi} - \frac{\pi}{3L^2}, & L = 0 \mod 4, \\ -\frac{2}{L \tan(\pi/L)} \simeq -\frac{2}{\pi} + \frac{2\pi}{3L^2}, & L = 2 \mod 4. \end{cases} \tag{B.6}$$

According to Eq. 4, this consequently leads to $c_0 = 1$ and $c_2 = -2$.

# C  Other CFT Hamiltonians with a $c_r < 0$ branch

Like the X-ZXZ model, there exist other 1D chains whose ground state energies, as a function of their length, have positive and negative $c$ branches. Here we provide two other examples of such models.

## C.1  Spin-1 chain

The model is a spin-1 chain with the Hamiltonian

$$H_{\text{Haldane-AFM}} = \sum_i \mathbf{S}_i \cdot \mathbf{S}_{i+1} + D \sum_i (S_i^z)^2. \tag{C.1}$$

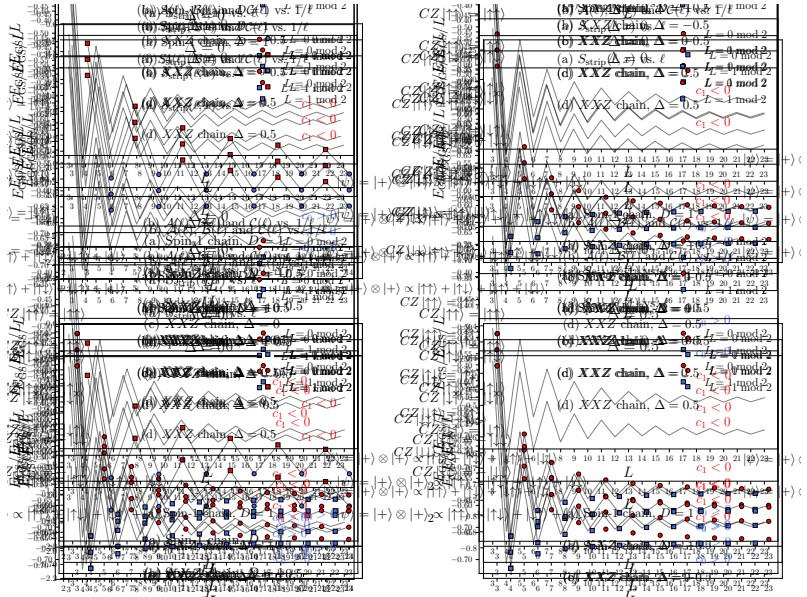

Figure 8: Energy density of different 1$D$ CFT chains with an even-odd effect. The energies are calculated through exact diagonalization. (a) Energy density of the spin-1 chain introduced in C.1 at $D = 1 \approx D_c$, (b-d) energy density of the $XXZ$ chain at different values of $\Delta$, namely $\Delta = -0.5, 0, 0.5$. It can be numerically confirmed that the system exhibits the even-odd effect at any values of $-1 < \Delta < 1$, where the system is in its gapless phase. However, the values of $c_0$ and $c_1$ are not universal and depend on $\Delta$.

This model undergoes a phase transition at $D_c \approx 1$ between a trivial paramagnetic phase for large $D$ and the Haldane phase for small $D$ [48, 77], which exhibits a $\mathbb{Z}_2 \times \mathbb{Z}_2$ SPT [78–80]. The critical point belongs to the same universality class as the X - ZXZ chain and is described by a $c = 1$ CFT [79]. As shown in Fig. S2 (a), the ground state energy of the system with periodic boundary conditions, has two branches of $c_0 > 0$ for even system sizes and $c_1 < 0$ for odd ones.

In order to realize the snake physics with this Hamiltonian, it is necessary to have a mod 4 effect instead of an even-odd effect. This can be accomplished by using two independent copies of the spin-1 chain. More precisely, the ground state energy of

$$H_{\text{Doubled Haldane-AFM}}(L) = \sum_{i=1}^{L} \mathbf{S}_i \cdot \mathbf{S}_{i+2} + D \sum_{i=1}^{L} (S_i^z)^2, \tag{C.2}$$

at the quantum critical point $D = D_c$ has $c_0 > 0$ for $L = 0$ mod 4 and $c_2 < 0$ for $L = 2$ mod 4.

## C.2 XXZ model

The spin-1/2 XXZ model with the Hamiltonian

$$H_{XXZ} = \frac{1}{2} \sum_i \left( \tau_i^x \tau_{i+1}^x + \tau_i^y \tau_{i+1}^y + \Delta \tau_i^z \tau_{i+1}^z \right), \tag{C.3}$$

is a c = 1 CFT for $-1 < \Delta \leq 1$ [81]. The $XXZ$ model in its critical phase exhibits an even-odd effect when it has periodic boundary conditions. The plot of energy density is provided in Fig. S2 (b-d) for $\Delta = -0.5$, $\Delta = 0.5$ and $\Delta = 0$ ($XX$ chain). Similar to the previous examples,

the doubled Hamiltonian,

$$H_{\text{Doubled XXZ}}(L) = \frac{1}{2} \sum_{i=1}^{L} \left( \tau_i^x \tau_{i+2}^x + \tau_i^y \tau_{i+2}^y + \Delta \tau_i^z \tau_{i+2}^z \right), \tag{C.4}$$

leads to a mod 4 effect.

In the special case of $\Delta = 0$ (XX model), the model can be solved exactly using the Jordan-Wigner transformation, allowing us to analytically calculate $c_0$ and $c_2$ as explained below. First, we determine the relationship between the Casimir coefficients of the single and double chain models. The doubled model consists of two independent chains, each of length $L/2$, so we have:

$$\frac{E_{\text{double}}(L)}{L} = \frac{2E_{\text{single}}(L/2)}{L} = \frac{2}{L} \left( \epsilon_0 L/2 - \frac{\pi c_{\text{single}}(L/2)}{L/2} + \cdots \right), \tag{C.5}$$

which leads to

$$c_{\text{double}}(L) = 4c_{\text{single}}(L/2). \tag{C.6}$$

Now, for a single $XX$ chain, the Jordan-Wigner transformation yields the ground state energy as

$$\frac{E_{\text{single}}(L)}{L} = \begin{cases} -\frac{2}{L \sin(\pi/L)} \simeq -\frac{2}{\pi} - \frac{\pi}{3L^2}, & L = 0 \bmod 2, \\ -\frac{\cos(\pi/L)}{L \sin(\pi/2L)} \simeq -\frac{2}{\pi} + \frac{11\pi}{12L^2}, & L = 1 \bmod 2. \end{cases} \tag{C.7}$$

Therefore, according to Eq. C.6, for the doubled $XX$ model, we find $c_0 = 4$ and $c_2 = -11$.

# D  Proof of $L = 2 \bmod 4$ for contractible loops in a fully packed configuration

In this appendix, we prove that contractible loops in a fully packed configuration cannot be of length $L = 0 \bmod 4$, and must thus be of length $L = 2 \bmod 4$ (since loop lengths are always even).

The proof is in two steps. First, we show that contractible loops with $L = 0 \bmod 4$ necessarily have an odd number of honeycomb vertices in their interior. Second, we show that an area of the honeycomb lattice with an odd number of vertices cannot host a fully packed configuration. By combining the two results, we conclude that a contractible domain wall with $L = 0 \bmod 4$ is not consistent with a fully packed configuration.

## D.1  First result

In this first result, we show that contractible loops with $L = 0 \bmod 4$ necessarily have an odd number of honeycomb vertices in their interior. A simple proof can be found in Ref. [82], which we reproduce here for convenience.

Let $L$ be the length of the loop, let $k$ be the number of hexagons inside the loop, and let $x$ be the number of vertices which are strictly inside the loop. Let $z$ denote the number of obtuse loop vertices (i.e. vertices at which the interior angle drawn by the loop is 120 degrees) and $y$ the number of reflex loop vertices (i.e. vertices at which the interior angle drawn by the loop is 240 degrees). If we orient the loop counterclockwise, the obtuse vertices correspond to left turns and the reflex vertices correspond to right turns. We know that $z + y = L$ since each loop vertex is either obtuse of reflex. We also know that $z - y = 6$ since the loop needs to close: it needs to do 6 more left turns than right turns.

Consider cutting each interior hexagon into 12 right triangles by cutting along all its axes of symmetry. There are two ways of counting the number $t$ of such triangles: $t = 12k$, but

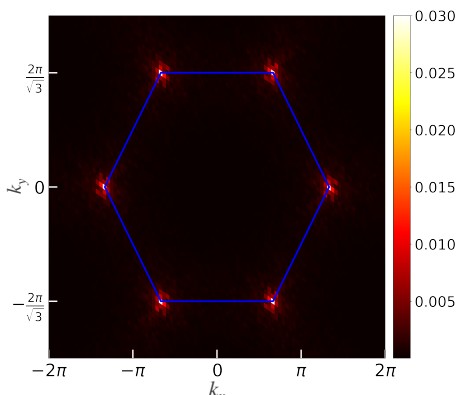

Figure 9: Color map of $C_{ZZ}(\mathbf{k})$ for an $\ell = 48$ system at $\tilde{T} = 4$ which peaks at the corners of the Brillouin zone.

also $t = 6x + 4y + 2z$. The first way is obtained by summing over hexagons. The second way is obtained by summing over interior points first, each of which is surrounded by 6 triangles. One should then add the triangles which touch the boundary: each obtuse vertex contributes two triangles, whereas each reflex vertex contributes four.

Finally, one finds $12k = t = 6x + 4y + 2z = 6x + 3(y + z) + (y - z) = 6x + 3L - 6$. Dividing by six gives $L/2 = 2k - x + 1$. This means $L = 2 \bmod 4$ if the number of interior points $x$ is even, and $L = 0 \bmod 4$ if the number of interior points $x$ is odd.

### D.2 Second result

A fully packed configuration must have a domain wall passing through each vertex of the honeycomb lattice. If $x$ is the number of vertices inside an area of the honeycomb lattice, a fully packed configuration in that area is a configuration of loops such that $x = \sum_i L_i$, where $i$ is the loop index. Since each $L_i$ is even, there can be no fully packed configuration if $x$ is odd.

## E  More details on the $\sigma^z \sigma^z$ correlator

### E.1 Numerical results

As usual for triangular lattice antiferromagnets, the spin correlation function $C_{ZZ}(\mathbf{x}) = \langle \sigma^z(0) \sigma^z(\mathbf{x}) \rangle$ has lattice scale oscillations corresponding to the K and K' wavevectors at the corner of the Brillouin zone. On top of these lattice scale oscillations, in the long distance limit, we expect the spin correlation $C_{ZZ}(\mathbf{x}) = \langle \sigma^z(0) \sigma^z(\mathbf{x}) \rangle$ to have a power law decaying envelope: $C_{ZZ}(\mathbf{x}) \sim |\mathbf{x}|^{-\eta}$. Assuming the structure factor defined by $C_{ZZ}(\mathbf{k}) \equiv 1/\ell^2 \sum_{\mathbf{k}} C_{ZZ}(\mathbf{x}) e^{-i\mathbf{k} \cdot \mathbf{x}}$ has a sharp peak at $\mathbf{k} = \mathbf{Q}$, we find $C_{ZZ}(\mathbf{Q}) \sim \ell^{-\eta}$. Fig. 9 shows that $\mathbf{Q}$ is indeed at the corners of the Brillouin zone. The value of $\eta$ can be obtained using the graph of $C_{ZZ}(\mathbf{Q})$ vs. $\ell$ (Fig. 3), which gives $\eta \approx 0.65$. In Appendix E.2, we propose a CFT argument based on Coulomb gas which predicts $\eta = 2/3$.

### E.2 CFT prediction

The $C_{ZZ}(r) = \langle \sigma^z(0) \sigma^z(r) \rangle \sim |r|^{-\eta}$ correlator can be interpreted as a correlator of twist operators in the CFT which flip the sign of the fugacity for loops which enclose one point (say 0) but not the other ($r$) [83]. Inspired by Ref. [83], we propose that the scaling dimension for

$\sigma^z$ is given by

$$\Delta_{\sigma^z}(n) = \frac{1}{2g(n)}(\chi'^2 - \chi^2), \tag{E.1}$$

where $g(n) = 1 - e_0(n)$ with $2\cos(\pi e_0) = n$, $\chi' = 2/3$ and $\chi = 1/3$. The correlation function is given by $C_{ZZ}(r) \sim 1/|r|^{2\Delta_z}$.

If we apply this to the case of $n = 1$, we find $g(n = 1) = 2/3$ and $\Delta(n = 1) = 1/4$, which reproduces the well-known $\langle \sigma^z(0)\sigma^z(r) \rangle \sim \cos(2\pi r/3)/\sqrt{|r|}$ dependence of spin correlation function in the Ising antiferromagnet on the triangular lattice [50].

For our case ($n = 0$), we find $\Delta(n = 0) = 1/3$, and thus $C_{ZZ}(r) \sim 1/|r|^{2/3}$, which is close to the exponent $\simeq 0.65$ we observed numerically.

## F  Worm Update for Monte Carlo algorithm

Fully-packed loop configurations on a hexagonal lattice can be ergodically sampled using a classical Monte Carlo algorithm with worm updates [54–56]. In such loop configurations, all vertices will generally touch an even number of occupied bonds (links). If $B$ is the complete set of occupied bonds constituting a given loop configuration, then the worm update is initiated by attaching a single bond to $B$ with vertices $a$ and $b$: $B' = B \cup ab$. In this case, $a$ and $b$ are referred to as vertex defects since they touch an odd number of occupied bonds, hence removing the bond configuration from the subspace of valid loop configurations. To return back to this subspace, one random bond whose boundary includes either $a$ or $b$ is added or removed to $B'$, thereby updating the pair of vertex defects, and this process is repeated until the two vertex defects coincide: $a = b$. In our case, this cluster update can then be accepted or rejected with standard Metropolis sampling according to the Boltzamann weight given by $e^{-F[\mathcal{L}]/T}$. This cluster update procedure is summarized in page 12 of [54].

Here, $F[\mathcal{L}]$ is the free energy of the domain wall configuration $\mathcal{L}$, which is the sum of the free energies of each loop: $F[\mathcal{L}] = \sum_{l \in \mathcal{L}} F_{1D}(L_l, T)$. In the low-$T$ limit, $F_{1D}(L, T)$ can be expanded as

$$F_{1D}(L, T) = E_{GS}(L) - TS_{GS} + \dots, \tag{F.1}$$

which according to B.6, for the $X - ZXZ$ chain will be

$$F_{1D}(L = 4k + 2, T) = L\left(\epsilon_0 + \frac{2\pi}{3L^2} - \frac{T}{L}\log(2) + \cdots\right), \tag{F.2}$$

$$F_{1D}(L = 4k, T) = L\left(\epsilon_0 - \frac{\pi}{3L^2} + \cdots\right). \tag{F.3}$$

The $\log(2)$ term appears due to the fact that the $L = 2 \mod 4$ chain is doubly degenerate. By dropping unimportant constant terms and rewriting the above relations in terms of $\tilde{T}$, we obtain

$$F_{1D}(L = 4k + 2, T)/T = \frac{L}{\tilde{T}}\left(\frac{2\pi\ell^2}{3L^2} - \frac{\tilde{T}}{L}\log(2) + \mathcal{O}(\ell^{-2})\right), \tag{F.4}$$

$$F_{1D}(L = 4k, T)/T = \frac{L}{\tilde{T}}\left(\frac{-\pi\ell^2}{3L^2} + \mathcal{O}(\ell^{-2})\right). \tag{F.5}$$

# G  Calculations of the entanglement

## G.1  Entanglement on the 1D chain

In order to calculate the entanglement of a given eigenstate $|\Psi[\mathcal{L}]\rangle$, we need to calculate the entanglement of the $\tau$ dofs on the snake. As show in Fig. 5, after unfolding the snake, the subsystem $A$ is mapped to a union of disjoint intervals on the 1D chain.

Following [58–60], Renyi entropies of disjoint intervals in a CFT can be computed using the following analytical formula

$$\mathrm{Tr}[\rho_A^n] = C_n^N \left| \frac{\prod_{i<j}(u_j - u_i)(v_j - v_i)}{\prod_{i,j}(v_j - u_i)} \right|^{2\Delta_n} \mathcal{F}_{N,n}(\mathbf{x}), \tag{G.1}$$

where each interval ($i$ running from 1 to $N$) is between $u_i$ and $v_i$, $C_n$ are non-universal constants, and $\mathcal{F}_{N,n}(\mathbf{x})$ is a model-dependent scaling function of all the invariant ratios which can be constructed out of all the $2N$ endpoints of the intervals. This formula is interpreted as the correlation function of twist operators inserted at each interval endpoint, where the twist operators have the universal scaling dimension $\Delta_n = (c/12)(n - 1/n)$.

In this work, we will limit ourselves to the dominant contributions to the entanglement in the limit of large perimeter $\ell_A$ of the subsystem, which will turn out to follow a "non-local area law". For this reason, we omit the contribution from $\mathcal{F}_{N,n}(\mathbf{x})$ because it only gives a term of order $\mathcal{O}(1)$. Further, for the strip geometry we will consider below, we will show in G.3 that the contribution from the $C_n^N$ factor leads to a simple, local contribution to the area law which is model-dependent and not particularly interesting. We thus also omit it in the following.

The $n^{\mathrm{th}}$ Renyi entropy, $S_n = -\frac{1}{n-1}\log\left(\mathrm{Tr}[\rho_A^n]\right)$, is then given by

$$S_n = -2\frac{\Delta_n}{n-1} \log \left| \frac{\prod_{i<j}(u_j - u_i)(v_j - v_i)}{\prod_{i,j}(v_j - u_i)} \right| + \cdots, \tag{G.2}$$

where the dots represent the non-universal contributions from $\mathcal{F}_{N,n}(\mathbf{x})$ and $C_n^N$ which we dropped. In particular, the von Neumann entanglement entropy is obtained by taking $n \to 1$, leading to

$$S = -\frac{c}{3} \log \left| \frac{\prod_{i<j}(u_j - u_i)(v_j - v_i)}{\prod_{i,j}(v_j - u_i)} \right| + \ldots \tag{G.3}$$

Also, since we will always work with a finite-length snake with periodic boundary conditions, one should use the following replacement in Eqs. G.2 and G.3:

$$|u_i - u_j| \to \left(\frac{L}{\pi}\right) \sin\left(\frac{\pi |u_i - u_j|}{L}\right), \tag{G.4}$$

and similarly for $|v_i - v_j|$ and $|u_i - v_j|$, where $L$ is the length of the chain.

To test the validity of Eq. G.2 for the $X - ZXZ$ chain in the case of multiple intervals, we used the DMRG method described in Section 6.2 of Ref. [60], which computes the second Renyi entropy by performing "twist" operations between two MPS states at the endpoints of each interval. Specifically, we compute $S_2$ for four equal partitions: $u_1, u_2 = 0, L/2$ and $v_1, v_2 = L/4, 3L/4$ where $L$ is the linear size of the chain. Fig. 10 compares the results from DMRG to Eq. G.2 with $n = 2$ and with the modification given in Eq. G.4. Indeed, we find good agreement between DMRG and the CFT prediction for sufficiently large $L$. This justifies the use of Eq. G.2 when calculating the entanglement for the snake phase.

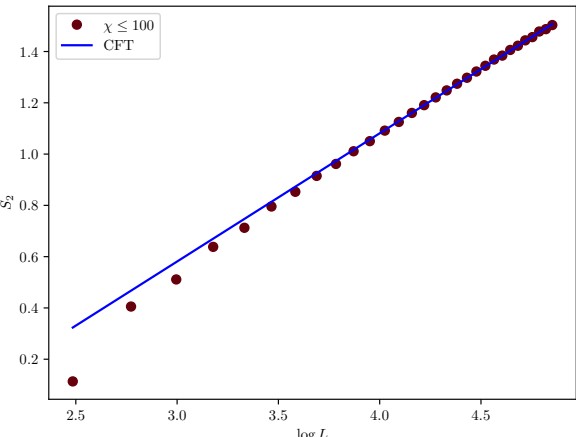

Figure 10: Second Renyi entropy for four equal partitions of the $X-ZXZ$ chain with linear size $L$, computed according to the DMRG method in Ref. [60] (with maximum bond dimension $\chi = 100$) and the analytical prediction for a CFT given in Eq. G.2. The entropies computed using DMRG are shifted by a constant value of $-1.889$ to account for the difference generated by the non-universal constant $c_2$ and scaling function $\mathcal{F}_{N,2}(\mathbf{x})$ given in Equation G.1.

## G.2 Averaging over snake configurations

Since the $\sigma$ degrees of freedom are classical, we are only interested in the quantum entanglement between $\tau$ degrees of freedom. We decide to average the value of the quantum entanglement over the thermally-generated $\sigma$ configurations. This provides us with the "quantum" contribution to the von Neumann entropy of the thermal density matrix after performing a partial trace over the $\tau$ degrees of freedom over the complement of the $A$ subsystem, as we now show.

The density matrix of the whole system is block diagonalized in the basis of domain wall configurations.

$$\rho = \frac{e^{-\beta H}}{\mathcal{Z}} = \begin{pmatrix} \frac{e^{-\beta H_1}}{\mathcal{Z}} & & \\ & \frac{e^{-\beta H_2}}{\mathcal{Z}} & \\ & & \ddots \end{pmatrix} = \begin{pmatrix} p_1 \rho_1 & & \\ & p_2 \rho_2 & \\ & & \ddots \end{pmatrix}, \tag{G.5}$$

where $\mathcal{Z}$ is the partition function, $p_i = \frac{\mathrm{Tr}(e^{-\beta H_i})}{\mathcal{Z}}$ is the thermal probability of being in the $i^{\mathrm{th}}$ domain wall configuration and $\rho_i = \frac{e^{-\beta H_i}}{\mathrm{Tr}(e^{-\beta H_i})}$ is the density matrix of the system in that configuration. By taking the partial trace over the $\tau$ spins in the subsystem $B$ (complement of $A$), the density matrix of the snake in subsystem $A$ can be expressed as

$$\rho^A = \begin{pmatrix} p_1 \rho_1^A & & \\ & p_2 \rho_2^A & \\ & & \ddots \end{pmatrix}. \tag{G.6}$$

The von Neumann entropy is obtained by the relation $S_A = -\mathrm{Tr}(\rho^A \log \rho^A)$ which gives

$$S_A = \sum_i \left[ -p_i \log p_i - p_i \mathrm{Tr}(\rho_i^A \log \rho_i^A) \right] \equiv S_{\mathrm{th}} + S_{\mathrm{qu}}, \tag{G.7}$$

where $S_{\text{th}} = -\sum_i p_i \log p_i$ is the thermal entropy of the domain wall configurations ($\sigma$ spins) and $S_{\text{qu}} = -\sum_i p_i \text{Tr}\big(\rho_i^A \log \rho_i^A\big)$ is the quantum entanglement of the $\tau$ spins, thermally averaged over snake configurations. The latter term is the quantum contribution which we are interested in. We simply call it the entanglement entropy and denote it by $S$ in the main text.

Practically, we calculate the average entanglement entropy over an ensemble of domain wall configurations generated by the Monte Carlo method discussed in Appendix. F. In order to find the entanglement entropy of each configuration, we add contributions of different domain walls calculated via Eq. G.3 and Eq. G.4. The only subtlety is that the definition of $u_i$ and $v_i$ is ambiguous on a lattice and needs to be regularized. We conventionally choose the following UV-regularization: first we number the sites on the loop from 1 to $L$. Then, for the $i^{\text{th}}$ interval, $u_i$ is defined as the first site of the interval placed inside the region $A$, and $v_i$ is defined as the first site which lies outside $A$. We also note that although in our calculations the temperature is not exactly zero, we use Eq. G.2 and Eq. G.4 and drop the finite-$T$ contribution which is parametrically small in system size and is not desired since we seek quantum contributions.

### G.3 Analytical prediction for the entanglement scaling

In this section, we derive a formula for the entanglement of a strip based on a Brownian motion model for the interval distribution (see Fig. 5 for the geometry of the strip and the definition of intervals $t_i$). As mentioned in the main text, since we only aim to reproduce the dominant scaling behavior of the entanglement with the partition size $\ell_A$, we assume it is sufficient to replace Eq. G.3 by $S = (c/3)\sum_{i=1}^{N_{\text{int}}} \log(t_i)$, where $t_i$ is the length of interval $i$. The average over snake configurations is then replaced by an average over interval length $\overline{f(t)} \equiv \int dt\, p(t) f(t)$ with distribution $p(t)$, and the entanglement reads $S = (c/3)N_{\text{int}}\overline{\log(t)} = (c/3)(2\ell_A/3)\overline{\log(t)}$.

There remains to find $p(t)$. In order to do this, let us neglect the correlations inherent to an $n = 0$ fully-packed self-avoiding walk and assume we are dealing with a "plain" random walk instead. The justification is that the geometrical exponent $\nu = 1/2$ for the $n = 0$ fully-packed self-avoiding walk is the same as that of a plain random walk. It might therefore be safe to neglect correlations and work with a random walk, as far as the qualitative behavior of $p(t)$ is concerned. As a reminder, the exponent $\nu$ relates the average end-to-end distance $|\mathbf{x}|$ of a walk after $t$ steps according to $|\mathbf{x}| \sim t^\nu$.

Using this random walk approximation, $t$ becomes the number of steps a random walk spends in partition $A$ before exiting, having started inside $A$ one lattice site away from the entanglement cut. This is known as a first passage time and was studied in the literature in a variety of geometries [84]. Let us consider an infinite cylinder of circumference $\ell$ and calculate the bipartite entanglement for a strip of width $x$. Taking the continuum limit, the random walk is described by a Brownian particle with probability distribution $U(X, Y, t)$:

$$\partial_t U = D\nabla^2 U\,, \tag{G.8}$$

with Dirichlet boundary conditions at the left and right entanglement cuts ($x = 0$ and $x = X$) and periodic boundary conditions along $Y$:

$$\begin{aligned} U(X = 0, Y, t) &= 0\,, \\ U(X = x, Y, t) &= 0\,, \\ U(X, Y + \ell, t) &= U(X, Y)\,, \end{aligned} \tag{G.9}$$

and with a Dirac delta initial condition located at $x = a$, which is just one lattice constant inside $A$ starting from the left cut ($a$ is the lattice constant):

$$U(X, Y, t = 0) = \delta(X - a)\delta(Y)\,. \tag{G.10}$$

We note that this is effectively a one-dimensional problem and one can completely forget about the $Y$ direction. Also, we define a dimensionless diffusion constant as $d = D\Delta t/a^2$, where $\Delta t$ is the time step for the random walk (we choose units such that $\Delta t = 1$ in the following).

Following standard procedure [84], the survival probability (i.e. the probability of the walker still being inside region $A$ at time $t$) is given by $S(t) = \int_0^x dX \int_0^{L_y} dY\, U(t)$ and the exit time distribution is given by $p(t) = -\frac{dS(t)}{dt}$.

After an elementary calculation, one finds

$$p_x(t) = -2D\pi \frac{1}{x^2}\vartheta_2'(z,q)\,, \tag{G.11}$$

with $\vartheta_2'(z,q)$ the derivative with respect to $z$ of the second elliptic theta function, $z = a\pi/x$ and $q = e^{-D(\pi/x)^2 4t}$. By using the Poisson resummation formula, we find a simple expression for $p(t)$ which is valid at times shorter than the Thouless time $t_{\mathrm{Th}} = x^2/D$:

$$p_x(t) \simeq \frac{1}{2}\frac{1}{\sqrt{\pi d}}\frac{1}{t^{3/2}}e^{-\frac{1}{4dt}} \quad (\text{for } t \ll x^2/D)\,. \tag{G.12}$$

We have thus recovered the Lévy distribution at short times. Beyond the Thouless time, the distribution decays exponentially:

$$p_x(t) \simeq 8D\pi^2 \frac{a}{x^3}e^{-D(\pi/x)^2 t} \quad (\text{for } t \gg x^2/D)\,. \tag{G.13}$$

We now want to calculate $\overline{\log(t)}_x \equiv \int dt\, p_x(t)\log(t)$ as a function of $x$. In the limit of $x \to \infty$, $p_x(t)$ is the Lévy distribution for all $t$, and we find the analytic expression $\overline{\log(t)}_{x\to\infty} = \gamma - \log(d)$ with $\gamma$ Euler's constant.

We have not found a closed form for $\overline{\log(t)}_x$ for general $x$, so we have evaluated it numerically (see Fig. 11). We find that, for $x \gg a$, it behaves as

$$\overline{\log t}_x \sim \overline{\log(t)}_{x\to\infty} - \frac{b}{x}\,, \tag{G.14}$$

with $b$ a constant of order 1 (for example, we extract $b \simeq 2.7732$ for $d = 1$). The prediction for the entanglement entropy in the regime $x \gg a$ is thus

$$S_{\mathrm{Strip}}(l,x) = 2\ell\left(A - \frac{B}{x}\right)\,, \tag{G.15}$$

with $A \propto \overline{\log(t)}_{x\to\infty}$ and $B \propto b$.

So far the calculation was done for a system which is infinitely long along $X$. In order to compare with numerics, we need to generalize this to the case of an $\ell$ by $\ell$ torus, in which case we know that, by symmetry, $S_{\mathrm{Strip}}(l,x)$ should be symmetric under $x \to \ell - x$. Inspired by Eq. G.4, we propose to do so by replacing $x$ by $\tilde{x} = \frac{\ell}{\pi}\sin\left(\frac{\pi x}{\ell}\right)$ in Eq. G.15. This finally leads to Eq. 9 in the main text.

Let us now comment on the non-universal contribution from $C_n^N$ in Eq. G.1 which we have dropped. This factor leads to a contribution to $S$ given by $\overline{N_{\mathrm{int}}}C_1'$, with $C_1' \equiv -\frac{dC_n}{dn}\Big|_{n=1}$ a model-dependent constant. By calculating numerically the entanglement $S_1$ of a single interval in the X-ZXZ chain, we found $C_1' \simeq 0.7 \pm 0.02$, which is close to the known value for the XX chain of $C_{1,\mathrm{XX}}' \simeq 0.726$ [85]. Further, we know that $\overline{N_{\mathrm{int}}}$ is a constant which is independent of $x$ and is equal to $2\ell/3$ in the thermodynamic limit. This can be shown by using the fact that fully-packed loop configurations have the following $U(1)$ symmetry: any two parallel straight lines along a principal axis of the triangular lattice cross the exact same number of loop strands.

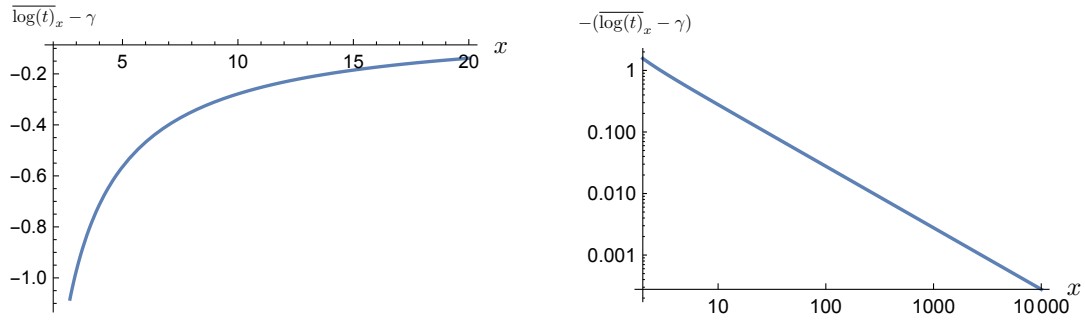

Figure 11: $\overline{\log t}_x - \overline{\log(t)}_{x \to \infty}$ vs $x$ in linear and log scales for a dimensionless diffusion constant $d = 1$. We find $\overline{\log t}_x \sim \gamma - \frac{b}{x}$ at large $x$ with $b \simeq 2.7732$.

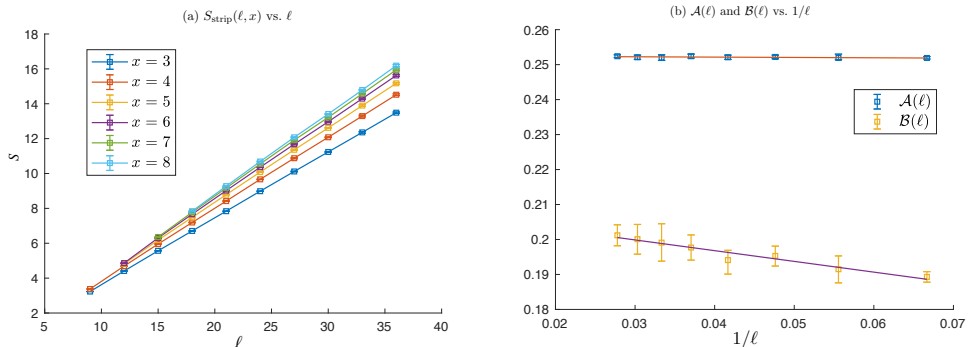

Figure 12: (a) Entanglement entropy of the strip shown in Fig. 5 vs. $\ell$ for fixed values of its width $x$. The graph shows that the entanglement entropy grows almost linearly, indicating that the leading contribution to the entropy follows the area law. (b) $\mathcal{A}(\ell)$ and $\mathcal{B}(\ell)$ along with their respective linear fits vs. $1/\ell$.

This means the number of loop strands crossing the entanglement cut is independent of $x$ for any given loop configuration. We also know that the number of intervals is simply the number of loop strands crossing the entanglement cuts divided by two. The snake configurations can thus be divided into different sectors based on their value for $N_{\text{int}}$. (For a discussion of these sectors, see [56]). Finally, we know that $\overline{N_{\text{int}}} = 2\ell/3$ (up to fluctuations which vanish in the thermodynamic limit) since the $N_{\text{int}} = 2\ell/3$ sector has the largest entropy [56]. All in all, this means we have dropped a contribution to $S$ which is equal to $(2\ell/3)C'_1$, which can be absorbed in the constant $A$ in Eq. G.15.

## G.4 More numerical results for the entanglement

More numerical results on the entanglement are provided in this appendix. First, Fig. S6 (a) explicitly illustrates the dominant area law scaling of the entanglement for the strip geometry. The dependence of the slope on $x$ reveals the non-local nature of the entanglement, as discussed in the main text.

Secondly, we show in Fig. S6 (b) the fitting parameters obtained by fitting Eq. G.15 to our numerical results, at fixed $\ell$ (see Fig. 6 in the main text for the fits). We denote the potentially $\ell$-dependent values obtained by these fits $\mathcal{A}(\ell)$ and $\mathcal{B}(\ell)$. We find a small drift of the fitting parameter $\mathcal{B}$ with $\ell$, and almost no drift for $\mathcal{A}$. Based on Fig. S6 (b), we propose a fit of the form

$$S_{\text{strip}}(\ell, x) = 2\ell\left(A - \frac{B}{\tilde{x}}\right) + 2\left(A' - \frac{B'}{\tilde{x}}\right) = 2\ell\left(\mathcal{A}(\ell) - \frac{\mathcal{B}(\ell)}{\tilde{x}}\right), \tag{G.16}$$

where $\mathcal{A}(\ell) = A + A'/\ell$, $\mathcal{B}(\ell) = B + B'/\ell$. We find $A = 0.2520 \pm 0.0002$, $A' \approx 0$, $B = 0.209 \pm 0.003$ and $B' = -0.31 \pm 0.07$.

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
