# Peer review of "Solvable models for 2+1D quantum critical points: Loop soups of 1+1D conformal field theories"

_SciPost Physics, doi:SciPost Phys. 16, 061 (2024)_

## Round 1 · Referee Report · Anonymous · 2023-12-10

Report
I read this interesting paper with great interest. The authors consider a system of two types of degrees of freedom: a system A which is an antiferromagnetic Ising model on the triangular lattice at zero temperature (a well known fully frustrated system with an extensive grind state degeneracy) and a system B which is to a quantum system of spin-1/2 spins defined o the dual (hexagonal) lattice. The authors study several models with the property that the two systems couple only across the domain walls of system A (which enumerate the macroscopic ground state degeneracy). In this fashion the domain walls become "decorated" by the degrees of freedom of system B. The result is an ensemble of domain walls with quantum mechanical degrees of freedom residing on them. They investigate the nature of the phases of this interesting system. They find that different types of local interactions lead to either a phase in which the loops ("decorated domain walls") are small (a "snake phase") or a phase in which they are long and for stripe like patterns. As far as I can tell the models that they considered the decorated loops do not interact with each other although are self-avoiding. However interactions between loops can alter the physics. For instance one can imagine that short-range interactions would lead to a crystal-type phase of short loops. Phases of this type were found in quantum dimer models which have some similarities with these models. I want to note that the arguments of the sign of the Casimir term are correct for loops which are much larger than the lattice spacing . In fact studies on the effects of boundary conditions in 1+1 dimensional CFTs by Cardy and others found that the sign could indeed be changed upon twisting the boundary conditions. The dependence of the coefficient on the length of loop is equivalent to a change of boundary conditions in the CFT. On the other hand, the Casimir term is not a well defined concept for finite loops. In this sense these arguments hold only in phases in which the loops are long. I am not sure how do the authors deal with the smaller loops, which are present. I want to comment that questions of this type were raise in the context of stripe phases resulting from doping a Mott insulator (e.g. in the work of Kivelson et al, Nature 393, 550, 1998) and later papers in the early 2000s on "smectic" or "sliding" Luttinger liquids.
In summary, I find this paper interesting and the material suitable for publication. However before I give my full endorsement I would like to know the response of the authors to the questions that I raised.

---

## Round 1 · Referee Report · Anonymous · 2023-12-15

Strengths
From the point of view of theoretical physics, creativity
Weaknesses
From the point of view of material science, connection to a material.
Report
This is a very well written and instructive paper that
I enjoyed reading. I recommend publication in SciPos essentially as is,
except for minor corrections.
The idea of this paper is to construct quantum-critical points in
(2+1)-dimensional spacetime as follows.
Step 1:
The nearest-neighbor ferromagnetic Ising Hamiltonian on the triangular lattice is the first ingredient. It is defined to be a sum of commuting projectors, one for each directed nearest-neighbor bond (direct bond)
of the triangular lattice, such that each projector selects with its
eigenvalue zero two parallel spins,
while it selects with its eigenvalue +1 two anti-parallel spins,
along the spin-1/2 quantization axis.
The ground states of this Hamiltonian are the twofold-degenerate
and gapped ferromagnetic ground states.
Step 2:
For each direct bond from the triangular lattice,
one defines a dual bond by demanding that the latter
is orthogonal to the former and they intersect through their midpoints.
The set of dual bonds defines the dual lattice, a honeycomb lattice.
Each site of the dual lattice hosts decoration degrees of freedom (dofs)
chosen to be spin-1/2 degrees of freedom.
Step 3:
Each projector from Step 1
is multiplied by a Hermitean polynomial
for the two dofs assigned to the dual bond attached to the direct bond
on which the projector from step 1 acts.
This is the quantum many-body Hamiltonian (1) defined in this paper.
Step 4:
Hamiltonian (1) can be rewritten as a sum over commuting Hermitean operators
$H_{1D}(l)$
labeled by the element $l$ belonging to the set $\mathcal{L}$
made of all non-crossing loops of the dual lattice,
whereby $H_{1D}(l)$ for each non-crossing loop $l$
is assigned the sum over the polynomials from step 3
for all the dual bonds whose union defines
the non-crossing loop $l$ from the dual lattice.
The subscript ``1D'' emphasizes that $H_{1D}(l)$ is defined on a
one-dimensional lattice made of $|l|$ sites, whereby
$|l|$ denotes the cardinality of the loop $l$.
Step 5:
It is assumed that
(i) the ground-state energy
$E_{1D;GS}(l)$ of $H_{1D}(l)$
is negative for any $l$ and
(ii) $H_{1D}(l)$ realizes a conformal field theory (CFT)
in the thermodynamic limit $|l|\to\infty$, i.e.,
$
\frac{E_{1D;GS}(l)}{|l|}=-|\epsilon_{0}|-\frac{\pi c_{l}}{3||l^{2}}
+\mathcal{|l|^{-3}}
$
Here, $c_{l}$ is a number of order one in powers of $|l|$
that can be negative for some values of $|l|\,\mathrm{mod}\, n$
with $n$ some given integer. The leading correction
$-\frac{\pi c_{l}}{3|l|^{2}}$
is called the Casimir energy.
The results from this paper are:
To leading order in the ground-state energy,
the ground state of Hamiltonian (1) is microscopically degenerate
as any loop covering of the dual lattice
that visits all sites of the dual lattice,
the so called subset of fully packed non-crossing loops from
$\mathcal{L}$, delivers a ground state.
When the Casimir energy is negative,
the degeneracy of the fully packed non-crossing loops is partially
lifted by selecting fully packed non-crossing loops made of
the shortest allowed closed loops. This gives as the ground state
an hexagonal solid that breaks spontaneously translation symmetry
and supports a gap.
When the Casimir energy is positive,
the degeneracy of the fully packed non-crossing loops is
partially lifted by selecting all non-crossing loop that
visits every sites (a snake). In the thermodynamic limit,
this gives rise to a (2+1)-dimensional quantum critical theory
with unusual scaling properties, as shown analytically
and numerical in the rest of the paper.
Several examples of
$H_{1D}(l)$
such that they can be tuned to (1+1)-dimensional criticality
with a positive Casimir energy are given.
In turn, the (2+1)-dimensional lattice Hamiltonian (1)
displays (2+1)-dimensional quantum criticality.
Zero temperature correlation functions are studied analytically and
numerically so as to extract scaling exponents. It is found that
the scaling exponent at the (2+1)-dimensional quantum critical
for the equal-time two-point correlation function of a
primary field from the (1+1)-dimensional quantum criticality
associated to $H_{1D}(l)$ is twice the value of the scaling exponent
of this primary field.
Finite temperature correlation functions are shown to be
sensitive to the order of limit $L\to\infty$ and $T\to0$ with $L$
the linear size of the triangular lattice and $T$ the temperature.
When $L\to\infty$ before $T\to0$,
$1/T$ plays the role of an infrared cutoff.
When $T\to0$ before $L\to\infty$, order by disorder selects
a stripe phase from $T>0$ to a critical temperature at which
a first-order transition into a snake phase takes place.
Whereas the two-point spatial correlation functions
of the primary fields of $H_{1D}(l)$
at the (2+1)-dimensional quantum critical point share
the same scaling exponents as those of $H_{1D}(l)$,
the bipartite entanglement of the two-dimensional soup of
(1+1)-dimensional CFTs obeys a non-local area law.
Comments:
1) This is a pure theory paper
in which a lattice Hamiltonian is proposed and studied.
The connection to the "real world'' is presented in the third
paragraph. Would the preprint
arXiv:2311.05004 [pdf, other] cond-mat.stat-mech
Fluctuation-induced spin nematic order in magnetic charge-ice
Authors: A. Hemmatzade, K. Essafi, M. Taillefumier, M. Müller,
T. Fennell, P. M. Derlet
not realize some three-dimensional version of this manuscript?
2) Why is there a summation over Ising configurations
on the right-hand side of Eq. (2)?
3) Typo:
A pair of parenthesis is missing on the right-hand side of Eq.\ (5).
4) In the first paragraph of the discussion of non-vanishing temperature,
there are three scaling relations, whereby the first two is supposed
to imply the third. Was there not a typo in one of the first two
scaling relations?
5) Typo:
"... the chain length module some integer, ..."
-> ^
"... the chain length modulo some integer, ..."
^

---

## Round 1 · Referee Report · Anonymous · 2023-12-21

Strengths
The manuscript presents an ingenious model in 2+1 dimensions, showing some excellent physics, which is understood in great depth using concepts ranging from quantum conformal field theories to the statistical physics of loops.
Weaknesses
The model is finely tuned, probably far from the real world.
Report
In the manuscript, the authors design a model of classical loops on the honeycomb lattice, where the loops are decorated with quantum degrees of freedom. Cleverly choosing the quantum Hamiltonian for the loop variables, they achieve longer loops to have lower ground state energy. At zero temperature, they end up with a manifold of long loops (a snake state) traversing the system. The quantum model for a loop is considered quantum-critical at a quantum phase transition between two gapped phases. The authors then study correlations of the decorated loops, find algebraic decay, and explain the measured exponents originating in the interplay of the quantum correlation and that of a self-avoiding walk. They also examine finite temperature behavior and quantum entanglement and provide a nice but involved argument for the area law they observed. It will be interesting to see what happens if the loops become quantum, with off-diagonal matrix elements reconfiguring them, like in the quantum dimer/ice model.
This is a nicely written manuscript, with careful numerics and clever theory (detailed explanations are given in appendices). I would like to recommend the acceptance for publication.
I only have a few primarily cosmetic comments listed below.
- In Eq. (1), I believe that the H_dec is symmetric, H_dec(1,2) = H_dec(2,1). It is worth mentioning explicitly.
- Regarding the style: It would be helpful if acronyms were in capitals, e.g., "dofs".
- In Fig 2, could the author also show the energies as a function of 1/L - it would be more informative.
- I would like to mention yet another paper with decorated critical loops (I leave it to the authors to decide whether to mention it): D. Poilblanc et al., Phys. Rev. B 75, 220503(R) (2007).

---

## Round 2 · Author Response

We are grateful to referees for their review of our paper. We addressed their comments in a point-by-point fashion below.
\noindent \underline{\bf Response to Anonymous Report 1} \medskip
I read this interesting paper with great interest. The authors consider a system of two types of degrees of freedom: a system A which is an antiferromagnetic Ising model on the triangular lattice at zero temperature (a well known fully frustrated system with an extensive grind state degeneracy) and a system B which is to a quantum system of spin-1/2 spins defined o the dual (hexagonal) lattice. The authors study several models with the property that the two systems couple only across the domain walls of system A (which enumerate the macroscopic ground state degeneracy). In this fashion the domain walls become "decorated" by the degrees of freedom of system B. The result is an ensemble of domain walls with quantum mechanical degrees of freedom residing on them. They investigate the nature of the phases of this interesting system. They find that different types of local interactions lead to either a phase in which the loops ("decorated domain walls") are small (a "snake phase") or a phase in which they are long and for stripe like patterns.
\response{ We thank the referee for her/his careful review and positive assessments of our work. Below, we respond to her/his questions one by one.}
As far as I can tell the models that they considered the decorated loops do not interact with each other although are self-avoiding. However interactions between loops can alter the physics. For instance one can imagine that short-range interactions would lead to a crystal-type phase of short loops. Phases of this type were found in quantum dimer models which have some similarities with these models.
\response{ As the referee mentioned correctly, in our proposed model, the interactions are effectively realized only along the loops. We also agree with the referee that introducing quantum fluctuations between different self-avoiding snake configurations can potentially lead to a different physics (this is also discussed in the discussion section of the manuscript). It is difficult to predict a priori what kind of phases these quantum fluctuations would stabilize without doing very heavy numerics, so we decided to leave it for future work. One way to approach it would be to use exact diagonalization to study the model with quantum fluctuations, following Pollmann et al. In their model, they had short loops before adding quantum fluctuations, and thus naturally found a plaquette phase of resonating short loops after adding them. In our case, since we start from a single long loop, it is not clear a priori what such a calculation would give and this is why we prefer not to speculate. We have added a sentence in the discussion to mention this.}
I want to note that the arguments of the sign of the Casimir term are correct for loops which are much larger than the lattice spacing . In fact studies on the effects of boundary conditions in 1+1 dimensional CFTs by Cardy and others found that the sign could indeed be changed upon twisting the boundary conditions. The dependence of the coefficient on the length of loop is equivalent to a change of boundary conditions in the CFT. On the other hand, the Casimir term is not a well defined concept for finite loops. In this sense these arguments hold only in phases in which the loops are long. I am not sure how do the authors deal with the smaller loops, which are present.
\response{ We agree with the referee that higher-order terms in $1/L$ for the energy of loops should appear for small $L$. However, these corrections are negligible for the models we have considered, which we have checked numerically. As one can see in Fig 2, the energy density agrees extremely well with the CFT prediction down to chains of length 6, which are the shortest possible loops on the honeycomb lattice. More generally, the only thing that matters in order to stabilize the snake phase is that the energy density for loops of length $L=4k+2$ is monotonically decreasing with $L$, which we have confirmed to be the case for all the models we have considered (see Fig 2 and Appendix C). We have added an explanation of this point in the legend of Fig 2 and in a footnote before Eq. 4}
I want to comment that questions of this type were raise in the context of stripe phases resulting from doping a Mott insulator (e.g. in the work of Kivelson et al, Nature 393, 550, 1998) and later papers in the early 2000s on "smectic" or "sliding" Luttinger liquids.
\response{We thank the referee for pointing out this very relevant part of the literature which we did not know about. We have added a sentence in the introduction discussing it, along with references to Nature 393, 550, 1998 and related works.}
In summary, I find this paper interesting and the material suitable for publication. However before I give my full endorsement I would like to know the response of the authors to the questions that I raised.
\response{We would like to thank again the referee for his/her interest in our work and his/her positive evaluation.}
\medskip \medskip \pagebreak \noindent \underline{\bf Response to Anonymous Report 2} \medskip
This is a very well written and instructive paper that I enjoyed reading. I recommend publication in SciPost essentially as is, except for minor corrections.
\response{We would like to express our sincere gratitude to the referee for their thoughtful review and positive evaluation of our manuscript. We appreciate his/her recommendation for publication of our work in Scipost Physics.}
The idea of this paper is to construct quantum-critical points in (2+1)-dimensional spacetime as follows.
Step 1: The nearest-neighbor ferromagnetic Ising Hamiltonian on the triangular lattice is the first ingredient. It is defined to be a sum of commuting projectors, one for each directed nearest-neighbor bond (direct bond) of the triangular lattice, such that each projector selects with its eigenvalue zero two parallel spins, while it selects with its eigenvalue +1 two anti-parallel spins, along the spin-1/2 quantization axis. The ground states of this Hamiltonian are the twofold-degenerate and gapped ferromagnetic ground states.
Step 2: For each direct bond from the triangular lattice, one defines a dual bond by demanding that the latter is orthogonal to the former and they intersect through their midpoints. The set of dual bonds defines the dual lattice, a honeycomb lattice. Each site of the dual lattice hosts decoration degrees of freedom (dofs) chosen to be spin-1/2 degrees of freedom.
Step 3: Each projector from Step 1 is multiplied by a Hermitean polynomial for the two dofs assigned to the dual bond attached to the direct bond on which the projector from step 1 acts. This is the quantum many-body Hamiltonian (1) defined in this paper.
Step 4: Hamiltonian (1) can be rewritten as a sum over commuting Hermitean operators $H_{\text{1D}}(l)$ labeled by the element $l$ belonging to the set $\mathcal{L}$ made of all non-crossing loops of the dual lattice, whereby $H_{\text{1D}}(l)$ for each non-crossing loop $l$ is assigned the sum over the polynomials from step 3 for all the dual bonds whose union defines the non-crossing loop $l$ from the dual lattice. The subscript 1D'' emphasizes that $H_{\text{1D}}(l)$ is defined on a one-dimensional lattice made of $|l|$ sites, whereby $|l|$ denotes the cardinality of the loop $l$.
Step 5: It is assumed that (i) the ground-state energy $E_{\text{1D,GS}}(l)$ of $H_{\text{1D}}(l)$ is negative for any $l$ and \ (ii) $H_{\text{1D}}(l)$ realizes a conformal field theory (CFT) in the thermodynamic limit $|l| \to \infty$, i.e., $E_{\text{1D,GS}}(l) = - |\epsilon_0| - \frac{\pi c_l}{3|l|^2} + |l|^{-3}$. Here, $c_l$ is a number of order one in powers of $|l|$ that can be negative for some values of $|l|$ mod $n$ with $n$ some given integer. The leading correction - $\frac{\pi c_l}{3|l|^2}$ is called the Casimir energy.
The results from this paper are:
To leading order in the ground-state energy, the ground state of Hamiltonian (1) is microscopically degenerate as any loop covering of the dual lattice that visits all sites of the dual lattice, the so called subset of fully packed non-crossing loops from $\mathcal{L}$, delivers a ground state.
When the Casimir energy is negative, the degeneracy of the fully packed non-crossing loops is partially lifted by selecting fully packed non-crossing loops made of the shortest allowed closed loops. This gives as the ground state an hexagonal solid that breaks spontaneously translation symmetry and supports a gap.
When the Casimir energy is positive, the degeneracy of the fully packed non-crossing loops is partially lifted by selecting all non-crossing loop that visits every sites (a snake). In the thermodynamic limit, this gives rise to a (2+1)-dimensional quantum critical theory with unusual scaling properties, as shown analytically and numerical in the rest of the paper.
Several examples of $H_{\text{1D}}(l)$ such that they can be tuned to (1+1)-dimensional criticality with a positive Casimir energy are given. In turn, the (2+1)-dimensional lattice Hamiltonian (1) displays (2+1)-dimensional quantum criticality.
Zero temperature correlation functions are studied analytically and numerically so as to extract scaling exponents. It is found that the scaling exponent at the (2+1)-dimensional quantum critical for the equal-time two-point correlation function of a primary field from the (1+1)-dimensional quantum criticality associated to $H_{\text{1D}}(l)$ is twice the value of the scaling exponent of this primary field.
Finite temperature correlation functions are shown to be sensitive to the order of limit $L \to \infty$ and $T \to \infty$ with $L$ the linear size of the triangular lattice and $T$ the temperature. When $L \to \infty$ before $T \to \infty$, $1/T$ plays the role of an infrared cutoff. When $T \to \infty$ before $L \to \infty$, order by disorder selects a stripe phase from $T > 0$ to a critical temperature at which a first-order transition into a snake phase takes place.
Whereas the two-point spatial correlation functions of the primary fields of $H_{\text{1D}}(l)$ at the (2+1)-dimensional quantum critical point share the same scaling exponents as those of $H_{\text{1D}}(l)$, the bipartite entanglement of the two-dimensional soup of (1+1)-dimensional CFTs obeys a non-local area law.
\response{We thank the referee again for carefully reading our manuscript and summarizing its main ideas and results.}
Comments:
1) This is a pure theory paper in which a lattice Hamiltonian is proposed and studied. The connection to the "real world'' is presented in the third paragraph. Would the preprint arXiv:2311.05004 [pdf, other] cond-mat.stat-mech Fluctuation-induced spin nematic order in magnetic charge-ice Authors: A. Hemmatzade, K. Essafi, M. Taillefumier, M. Müller, T. Fennell, P. M. Derlet not realize some three-dimensional version of this manuscript?
\response{Indeed, this is a 3D example of a charge-ice system realizing a model of loops decorated with gapless degrees of freedom. We have added a reference to it when we mention charge-ice in the introduction. We thank the referee for pointing out this work to us.}
2) Why is there a summation over Ising configurations on the right-hand side of Eq. (2)?
\response{The full Hamiltonian is defined on a Hilbert space of both Ising spins and decoration degrees of freedom. The Hamiltonian is block diagonal in the computational basis of Ising spin configurations ($\sigma$ spins). In order to define the full Hamiltonian, we thus need to sum over each block, which is done by the sum over Ising configurations. }
3) Typo: A pair of parenthesis is missing on the right-hand side of Eq.\ (5).
\response{We thank the author for bringing the typo in Eq. (5) to our attention. The corrections have been made in the updated version of the manuscript. }
4) In the first paragraph of the discussion of non-vanishing temperature, there are three scaling relations, whereby the first two is supposed to imply the third. Was there not a typo in one of the first two scaling relations?
\response{We extend our gratitude to the referee once more for identifying the typo. As the average number of domain walls, $\langle n_{\text{DW}} \rangle$, is proportional to $T \ell^2$, the second scaling relation needs to be adjusted to $\langle n_{\text{DW}} \rangle \sim \ell^2/\beta$.}
5) Typo: "... the chain length module some integer, ..." -> \ "... the chain length modulo some integer, ..."
\response{We appreciate the thorough review by the referee and identifying the typos. This has also been corrected in the revised manuscript.}
\medskip \medskip \pagebreak \noindent \underline{\bf Response to Anonymous Report 3} \medskip
In the manuscript, the authors design a model of classical loops on the honeycomb lattice, where the loops are decorated with quantum degrees of freedom. Cleverly choosing the quantum Hamiltonian for the loop variables, they achieve longer loops to have lower ground state energy. At zero temperature, they end up with a manifold of long loops (a snake state) traversing the system. The quantum model for a loop is considered quantum-critical at a quantum phase transition between two gapped phases. The authors then study correlations of the decorated loops, find algebraic decay, and explain the measured exponents originating in the interplay of the quantum correlation and that of a self-avoiding walk. They also examine finite temperature behavior and quantum entanglement and provide a nice but involved argument for the area law they observed. It will be interesting to see what happens if the loops become quantum, with off-diagonal matrix elements reconfiguring them, like in the quantum dimer/ice model. This is a nicely written manuscript, with careful numerics and clever theory (detailed explanations are given in appendices). I would like to recommend the acceptance for publication.
\response{We are delighted to learn that the referee found the manuscript well-written, with careful numerics and clever theory. We appreciate his/her positive assessment of the manuscript and are honored by his/her recommendation for acceptance for publication.}
I only have a few primarily cosmetic comments listed below.
- In Eq. (1), I believe that the H dec is symmetric, H dec(1,2) = H dec(2,1). It is worth mentioning explicitly.
\response{We would like to bring footnote 39 to the referee's attention in which we discuss this explicitly.}
- Regarding the style: It would be helpful if acronyms were in capitals, e.g., "dofs".
\response{We thank the referee for his/her feedback on the manuscript style. In the updated version of the manuscript, all acronyms have been capitalized.}
- In Fig 2, could the author also show the energies as a function of 1/L - it would be more informative.
\response{We appreciate the referee's suggestion to represent the energy in an alternative scale to enhance the clarity of the presentation. We have done so by updating Fig 2 to show $E_{\text{GS}}/L - \epsilon_0$ against $1/L^2$. This choice leads to two linear branches with positive and negative slopes for the $X-ZXZ$ chain, providing a more informative representation of the data.}
- I would like to mention yet another paper with decorated critical loops (I leave it to the authors to decide whether to mention it): D. Poilblanc et al., Phys. Rev. B 75, 220503(R) (2007).
\response{We thank the referee for pointing out this work to us which we did not know about. We have added a reference to it in the introduction.}

---

## Round 2 · List of Changes

A. Summary of Changes 1. Following Ref 1’s comments, we have added a comment on the role of quantum fluctuations in the discussion 2. Following Ref 1’s comments, we have added a comment on the validity of the CFT formula for the Casimir energy for short loops 3. Following Ref 1’s comments, we have added a mention of previous work on stripe phases in doped Mott insulators, along with references. 4. Following referee 2’s comments, we have corrected typos and added a reference. 5. In response to referee 3’s comment 2, we have capitalized acronyms. 6. In response to referee 3’s comment 3, we have modified Fig 2 and plotted the energy values in a scale where the graph is linear. 7. In response to referee 3’s comments, we have added a reference to Poilblanc et al.

---

## Editorial Decision

published